# Efficient Biological Data Acquisition through Inference Set Design

**Ihor Neporozhnii**[* 1, 2]   **Julien Roy**[*1]   **Emmanuel Bengio**[1]   **Jason Hartford**[1,3]
[1]Valence Labs   [2]University of Toronto   [3]University of Manchester
ihor.neporozhnii@mail.utoronto.ca & julien.roy@valencelabs.com

## Abstract

In drug discovery, highly automated high-throughput laboratories are used to screen a large number of compounds in search of effective drugs. These experiments are expensive, so one might hope to reduce their cost by only experimenting on a subset of the compounds, and predicting the outcomes of the remaining experiments. In this work, we model this scenario as a sequential subset selection problem: we aim to select the smallest set of candidates in order to achieve some desired level of accuracy for the system as a whole. Our key observation is that, if there is heterogeneity in the difficulty of the prediction problem across the input space, selectively obtaining the labels for the hardest examples in the acquisition pool will leave only the relatively easy examples to remain in the inference set, leading to better overall system performance. We call this mechanism *inference set design*, and propose the use of a confidence-based active learning solution to prune out these challenging examples. Our algorithm includes an explicit stopping criterion that interrupts the acquisition loop when it is sufficiently confident that the system has reached the target performance. Our empirical studies on image and molecular datasets, as well as a real-world large-scale biological assay, show that active learning for inference set design leads to significant reduction in experimental cost while retaining high system performance.

## 1 Introduction

High-throughput screening (HTS) laboratories have enabled scientists to efficiently perform whole-genome CRISPR knockouts and screen large compound libraries to discover effective therapeutics (Mayr & Bojanic, 2009; Wildey et al., 2017; Blay et al., 2020; Tom et al., 2024; Fay et al., 2023). However, conducting such experiments on every compound or gene in these vast design spaces remains resource-intensive. With typical screening libraries holding on the order of $10^5$ to $10^6$ compounds (Hughes et al., 2011) and the number of possible small molecules estimated at $10^{60}$ (Bohacek et al., 1996), the disparity between our screening capabilities and all which we *could* explore is staggering. Reducing experimental costs without compromising the quality of the generated data would allow to greatly accelerate biological and pharmaceutical research.

At any stage of a screening procedure, to reduce experimental costs, we can train a machine learning model on the data that has already been collected, and then use that model to predict experimental outcomes for the remainder of the experiments (Naik et al., 2013; Reker & Schneider, 2015; Dara et al., 2022). This procedure generates a *hybrid screen* of the target library, with some outcomes experimentally observed and others obtained from model predictions. However this approach entails interrelated questions: Which subset of the library should we use to maximize the accuracy of the predictions? How do we select this subset without access to all experimental outcomes? How do we ensure that we acquire a large enough collection to meet our accuracy goal?

This setting is similar to an active learning problem in that we want to select examples that maximize prediction accuracy, but instead of aiming to minimize generalization error on the entire data space, we focus solely on prediction on one particular set of experiments. The fact that this library is *finite* introduces an important difference: here the learner can influence the set of examples on which it

---

*Denotes equal contribution

is evaluated by strategically selecting the examples to be acquired. Since it is often the case that some experiments outcomes are inherently harder to predict than others, either due to their complex properties (Bengio et al., 2009), because of the partial observability of the system (Saleh et al., 2021; Krenn et al., 2020) or due to noise in the labeling function (Frénay & Verleysen, 2013; Lukasik et al., 2020), the learner can select these harder examples into the training set early on to avoid having to predict their outcomes at a later stage. Conversely, if a group of examples can be reliably predicted by the model, we can save experimental costs by *not* including them in the training set. We call this mechanism through which a learner can influence its own evaluation set: *inference set design*.

In this work, we propose an active learning-based solution to hybrid screening. Our approach uses the model's confidence to guide the selection of experiments and leverages the mechanism of inference set design to improve the system's performance on the target set. Our algorithm includes a practical stopping criterion that terminates the acquisition of additional labels once a lower bound on a target accuracy is exceeded, and we show that this bound provides probabilistic guarantee on the performance of the algorithm as long as the model is weakly calibrated. To validate our method, we conduct a series of empirical studies on image and molecular datasets, as well as a real-world case study in drug discovery. The results demonstrate that inference set design significantly reduces experimental costs while improving overall system performance. Importantly, this is true even when the generalization benefits of active learning-based acquisition functions are marginal compared to random search. This has important practical implications for active learning: if a problem is a hybrid screen—in the sense that one only needs good performance on a fixed, finite set of experiments— then evaluating generalization error dramatically understates the benefits of active learning. By combining simple active learning acquisition functions with an appropriate stopping criterion, we show that it is possible to make large scale screening far more efficient.

## 2 METHODS

In this section, we present the problem of efficiently evaluating a large, *finite* set of experimental designs, such as screening a compound library. We formalize it as a subset selection problem, and show that it can be solved using an active learning strategy. Active learning is effective in this setting because it essentially selects its own inference set; we call this mechanism, *inference set design*. We propose a stopping criterion to monitor the model's performance and trigger termination of the acquisition process and show that this algorithm allows to reach provably high levels of performance on a target set of samples. A pseudo-code is available in Appendix B.

### 2.1 REDUCING EXPERIMENTAL COSTS VIA HYBRID SCREENS AND INFERENCE SET DESIGN

We are motivated by the problem of efficient data acquisition of compound libraries for drug discovery applications. Each "experimental readout" (or label) $y$ requires us to run an experiment, which has some associated cost. We assume that we have a predefined *target set* $\mathcal{X}_{\text{target}} = \{x_i\}_{i=1}^{N_{\text{target}}}$ of experimental designs for which we want to acquire a readout, where $N_{\text{target}}$ represents the number of samples we wish to obtain labels or predictions for. For example, in drug discovery, a common set of experimental designs, $\mathcal{X}_{\text{target}}$, would be a library of thousands or even millions of compounds, each required to be tested on several cell types and at different concentrations. For each $x_i \in \mathcal{X}_{\text{target}}$, the associated experimental readout, $y_i$, would correspond to the drug's effect on the corresponding cell type. One way to reduce the acquisition costs of screening the entire target set is to train a predictive model $\hat{p}(x)$ on the *observation set* – the subset of the target set, $\mathcal{X}_{\text{obs}} \subseteq \mathcal{X}_{\text{target}}$, for which we already have observed the labels $y$. We can then use this model's predictions in place of the real labels on the remaining samples. Because experiments are typically run in multiple rounds, at each round $t$, the target set $\mathcal{X}_{\text{target}}$ can be partitioned into two mutually exclusive subsets:

- The *observation set* $\mathcal{X}_{\text{obs}}^t$, which contains the union of all experimental designs that have already been tested and for which the readouts, $\mathcal{Y}_{\text{obs}}^t$, have been observed. We denote their pairing the observation *dataset* $\mathcal{D}_{\text{obs}}^t := (\mathcal{X}, \mathcal{Y})_{\text{obs}}^t := \{(x_i, y_i)\}_{i=1}^{N_{\text{obs}}}$. At each step $t$, a constant number of $N_b$ samples are acquired and added to the observation set, causing this set to grow over time.
- The *inference set* $\mathcal{X}_{\text{inf}}^t := \mathcal{X}_{\text{target}} \setminus \mathcal{X}_{\text{obs}}^t$, which consists of the remaining experimental designs in the target set that have not yet been tested. This set shrinks over time as samples are selected from it for acquisition and transferred into the observation set.

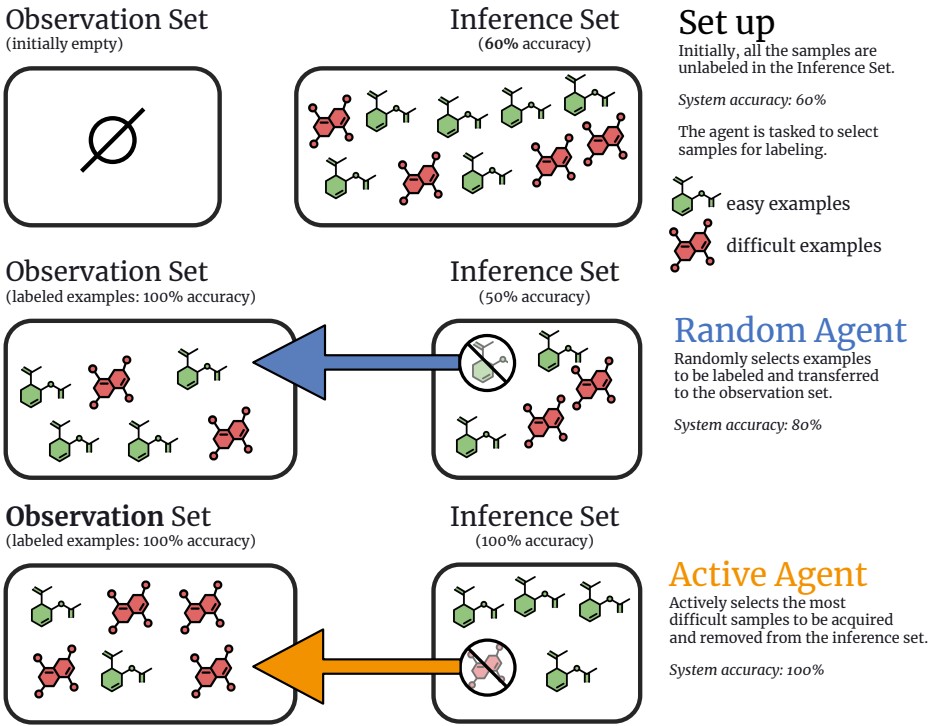

Figure 1: **Hybrid screen employing active learning as an inference set design strategy.** The goal is to produce a "hybrid dataset", composed of the labels collected in the observation set and the predictions of the model made on the inference set. The system's performance is reported as the "system accuracy", measured as a combination of observed labels on the observation set and of predictions made on the inference set. As samples are acquired and added to the observation set, the contribution of the accuracy of the prediction model gradually decreases. The aim of the active agent is to select informative examples while pruning out the most difficult ones in order to reach very high system accuracy without having to acquire all the samples.

Once acquisition is stopped, at timestep $t = \tau$, the output of the system is a *hybrid dataset*, i.e. the combination $(\mathcal{X}, \mathcal{Y})_{\text{obs}}^{\tau} \cup (\mathcal{X}, \hat{\mathcal{Y}})_{\text{inf}}^{\tau}$ of both the labeled pairs from the observation set and the predicted pairs from the inference set. We call this procedure a *hybrid screen*, as it provides a hybrid set of labels for the entire target set – some measured, some predicted. The system's performance at each step $t$ is evaluated based on its accuracy $\mu_{\text{sys}}^{t}$ across the entire target set, which consists of perfectly accurate labels $y$ from the observation set and the inferred labels $\hat{y}$ for the inference set:[1]

$$\mu_{\text{sys}}^{t} := \frac{1}{|\mathcal{X}_{\text{target}}|} \sum_{x_i \in \mathcal{X}_{\text{target}}} \mathbf{1}(x_i \in \mathcal{X}_{\text{obs}}^{t}) + \mathbf{1}(\hat{y}_i = y_i \mid x_i \in \mathcal{X}_{\text{inf}}^{t}) \tag{1}$$

In this formulation, acquiring all the labels would yield 100% accuracy, but at significant expense. We instead seek a system that reaches an accuracy of at least $\gamma$, while incurring minimal experimental costs. This can be formulated as the following combinatorial optimization problem,

$$\min_{\mathcal{X}_{\text{obs}} \subseteq \mathcal{X}_{\text{target}}} |\mathcal{X}_{\text{obs}}| \quad \text{such that} \quad \mu_{\text{sys}}^{t=\tau} > \gamma \tag{2}$$

where $|\mathcal{X}_{\text{obs}}|$ denotes the number of samples in the observation set. A natural approach is to treat this as an active learning problem. Like standard active learning, we can sequentially select subsets of data for labeling and the system accuracy will benefit from any improvements in generalization that arise from better selection of training examples. However, this setting has an important difference: we only ever evaluate $\hat{p}$ on $\mathcal{X}_{\text{inf}}$, and this set is under the control of the learner because any example that is selected for labeling is removed from the inference set. As a result, by choosing acquisition functions that select harder training examples first, we make the test-time task of the learner easier, as

---

[1]Equation 1 assumes a classification problem, but we also explore continuous outcomes $y$ in Appendix E.1.

the harder examples are effectively *pruned out* of the inference set. We call this approach *inference set design* – by selective label acquisition, the agent actively designs the *composition* of the inference set such that it can excel on it. This mechanism is illustrated in Figure 1.

The data selection heuristics for inference set design are similar to standard active learning, but there are two important differences between the settings. Firstly, if we evaluate performance under the standard active learning objective—generalization error measured on a held-out test set—we risk dramatically understating the performance improvement that result from inference set design. Instead, evaluating on the target set of interest allows to capture dramatical improvements over a random sampler even when only marginal improvements in generalization error can be observed. Secondly, unlike active learning, inference set design requires an explicit stopping criterion in order to decide when you have collected enough samples to meet the accuracy threshold, which we address in the following section.

## 2.2 EMPIRICAL STOPPING CRITERION

The key technical challenge of inference set design is deciding when to stop running new experiments. In a deployment environment, the labels for the inference set will remain unavailable, so we need to estimate the system performance from the collected data only. To avoid unnecessary experimentation, inference set design algorithms must employ an efficient stopping criterion. We address this by maintaining a probabilistic lower bound on the system accuracy, and stopping once this bound exceeds the critical threshold, $\gamma$. A simple approach to maintaining such a bound is to leverage the feedback from each round of experimentation.

We consider a multi-class classification problem. Formally, let $\mathcal{K} := \{1, 2, \ldots, K\}$ be the set of $K$ labels and $\hat{p}(x) := (\hat{p}_1(x), \ldots, \hat{p}_K(x))$ be the predicted probabilities for $x$ that $y = k$ for all $k \in \mathcal{K}$. We denote $y_i$ the true label for a sample $x_i$, $\hat{y}_i := \arg\max_k \hat{p}_k(x_i)$ the predicted class by the model and $\hat{v}_i := \max_k \hat{p}_k(x_i)$ the predicted probability value for that class. We consider the use of the commonly employed least-confidence acquisition function:

$$x_{\text{selected}} := \arg\max_{x_i \in \mathcal{X}_{\text{inf}} \setminus \mathcal{X}_b} \left(1 - \hat{v}_i\right), \quad \text{for} \quad i = 1, \ldots, N_b \tag{3}$$

where $N_b$ is the number of samples to be acquired, i.e. the acquisition batch-size. The samples $x_i \in \mathcal{X}_b$ are selected one by one by applying the least-confidence criterion in Equation 3 iteratively to the remaining samples in $\mathcal{X}_{\text{inf}}$ that have not yet been transferred to $\mathcal{X}_b$. Because at every round we select examples that the model regards as challenging, if the model maintains a correct calibration ordering, the performance on a batch $\mathcal{X}_b^t$ that we select at round $t$ should lower bound the performance on the entire inference set. We express this formally in the following lemma:

**Lemma 1.** *Assuming that $\hat{p}$'s top-predictions are weakly calibrated on the inference set, i.e. that for any two samples $x_1$ and $x_2$ in $\mathcal{D}_{inf}^{t-1}$, $\hat{v}_1 \geq \hat{v}_2 \implies P_{\mathcal{D}_{inf}^{t-1}}[Y = \hat{Y} | \hat{v}_1] \geq P_{\mathcal{D}_{inf}^{t-1}}[Y = \hat{Y} | \hat{v}_2]$, then the expected accuracy on a batch $\mathcal{D}_b^t := \{(x_i, y_i)\}_{i=1}^{N_b}$ selected by least-confidence from $\mathcal{X}_{inf}^{t-1}$ is less or equal than the expected accuracy on the remaining inference set: $P_{\mathcal{D}_b^t}[Y = \hat{Y}] \leq P_{\mathcal{D}_{inf}^t}[Y = \hat{Y}]$.*

See Appendix A.1 for more details and proof. Lemma 1 shows that although the samples in $\mathcal{X}_b^t$ are not independent and identically distributed samples from $\mathcal{X}_{\text{inf}}^t$, we can safely use the performance on $\mathcal{D}_b^t$ as a conservative proxy for performance on $\mathcal{D}_{\text{inf}}^t$. We thus leverage the *measured* performance on the acquired batch to estimate the performance on the remaining, *unobserved* inference set.

We can now use this conservative estimate to design an appropriate stopping criterion. Let $\hat{\mu}_b^t := \frac{1}{N_b} \sum_{i=1}^{N_b} \mathbf{1}(y_i = \hat{y}_i)$ denote the measured accuracy on the batch $\mathcal{D}_b^t$ at acquisition step $t$. A simple stopping criterion could consist in picking an accuracy threshold $\gamma$ for the desired system accuracy, and to stop acquiring new samples once our system accuracy estimate $\hat{\mu}_{\text{sys}}^t$ based on $\hat{\mu}_b^t$ surpasses $\gamma$:

$$\tau = \arg\min_{t \in 1, \ldots, T} \hat{\mu}_{\text{sys}}^t > \gamma \quad, \quad \hat{\mu}_{\text{sys}}^t = \frac{|\mathcal{X}_{\text{obs}}^t| + \hat{\mu}_b^t |\mathcal{X}_{\text{inf}}^t|}{|\mathcal{X}_{\text{target}}|} \tag{4}$$

where $\tau$ denotes the stopping time. Optionally, we could augment this criterion with a *patience* hyperparameter to make it more robust to the stochasticity introduced by the finite sample size $N_b$. A more principled approach however is to directly account for the statistical uncertainty around $\hat{\mu}_b^t$

by using it to derive a bound $\alpha_t$ on the true inference accuracy $\mu_{\text{inf}}^t$. Using standard concentration inequalities (see detailed derivation in Appendix A.2) we have that,

$$P(\mu_{\text{inf}}^t \leq \alpha_t) \leq \delta \quad \text{with} \quad \alpha_t = \min \left\{ a \in [0, \hat{\mu}_b^t] : \text{KL}(\hat{\mu}_b^t || a) \leq \frac{\log\left(\frac{1}{\delta}\right)}{N_b} \right\} \tag{5}$$

where $\text{KL}(\cdot || \cdot)$ denotes the Kullback-Leibler divergence and $\delta$ is the failure probability of the bound. We can thus use the lower-bound value $\alpha_t$ in place of $\hat{\mu}_b^t$ in Equation 4 to ensure achieving our desired system accuracy threshold $\gamma$ with probability at least $1 - \delta$.

To summarize, in this section, we presented a framework for hybrid screens leveraging active learning to perform inference set design by selectively acquiring labels for hard-to-predict examples, thereby simplifying the inference set for easier prediction, and using a conservative stopping criterion to ensure that the desired level of system accuracy is achieved at stopping time. The pseudocode for our algorithm is presented in Appendix B.

## 3 RELATED WORK

In this work, we apply Active Learning (AL) to the problem of hybrid screens for biological data acquisition. Warmuth et al. (2003) were among the first to apply AL to the field of drug discovery, with an acquisition function based on support vector machines for binding affinity predictors. AL has since then been explored as a tool for problems such as virtual screening (Fujiwara et al., 2008), cancer monitoring (Danziger et al., 2009), protein-protein interaction (Mohamed et al., 2010), compound classification (Lang et al., 2016) and chemical space exploration (Smith et al., 2018).

These approaches seek to produce a better model than training on random samples would, and are to be distinguished from bayesian optimization methods (Graff et al., 2021; Gorantla et al., 2024), which also fall under the umbrella of "active learning" but instead use the model to seek specific samples (e.g. with high binding affinity) and often focus on addressing the exploration-exploitation dilemma which has been widely studied by bandit algorithms (Svensson et al., 2022). For hybrid screens, we seek a model that can accurately predict the labels of *all* samples in the inference set, without attempting to maximize the value of the readouts. Our setting, focusing on a fixed set of data, is sometimes referred to as pool-based active learning (Wu, 2019; Zhan et al., 2021). Another similar approach called transductive experimental design aims to select samples that improve the predictions for a specific pre-defined test set (Yu et al., 2006; Hübotter et al., 2024). A key difference is that most active learning work (including pool-based AL and transductive experimental design) tends to focus evaluations on a held-out test set (Atlas et al., 1989; Cohn et al., 1994; Gal et al., 2017; Margatina et al., 2021; Zhan et al., 2021; Luo et al., 2023; Li et al., 2024) or restrict the algorithm to very limited acquisition budgets (e.g. only 10% of the data pool) (Yu et al., 2006; Wu, 2019), which generally does not leave enough margin for mechanisms such as inference set design to dominate.

A closely related problem is that of core set selection (Plutowski & White, 1993; Bachem et al., 2017; Guo et al., 2022), where the goal is to select the smallest set of samples from a pool of data such that a model trained on this subset performs on par with a model trained on the entire pool. The closest works to ours are that of Sener & Savarese (2017) and Li & Rangarajan (2019), which make an explicit connection between active learning and core set selection and also seek to improve performance on the unselected examples. In our work however, we focus on uncovering the key mechanism in action, inference set design, and show empirical validations across a wide variety of academic and real-life datasets.

## 4 EXPERIMENTS

In our experiments[2], we aim at highlighting the key role that inference set design plays in allowing AL agents to reach high levels of performance on the inference set (Section 4.1), validating the robustness of this mechanism on molecular datasets used for drug discovery (Section 4.2), applying this method to a challenging real-world case of compound library screening on a large scale

---

[2]The code is available at https://github.com/ineporozhnii/inference_set_design. All datasets to reproduce our results are publicly available, except one proprietary dataset for the results in Figure 8.

phenotypic assays (Section 4.3) and empirically validating our theoretical assumption and stopping criterion (Section 4.4). Unless specified otherwise, all curves show average metrics across 3 seeds and the shaded areas denote the standard error of the mean. Vertical lines denote the average stopping time, with min and max intervals (left to right). For all modalities, we use vectorized representations as input and our model $\hat{p}$ is parameterized using a feed-forward deep neural network with residual connections. Additional details regarding data pre-processing, network architectures and hyperparameters are presented in Appendices C and D.

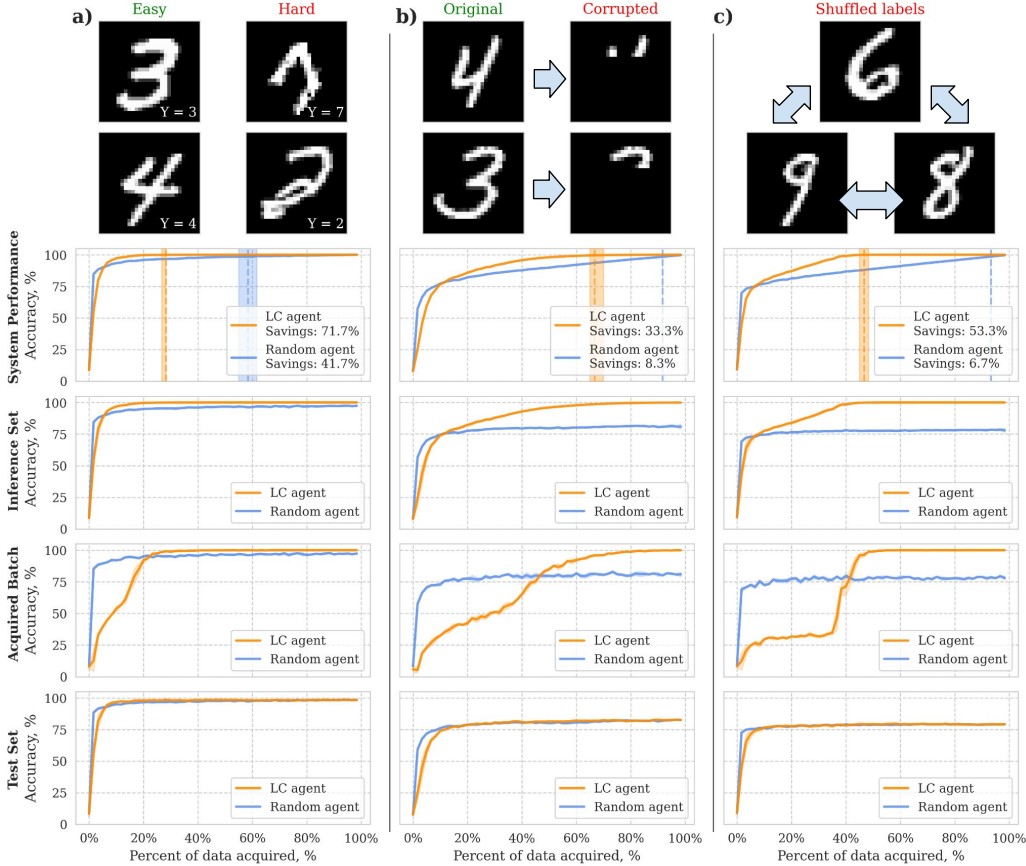

Figure 2: Performance of least-confidence (LC) and random acquisition functions on variations of MNIST (columns) across 3 seeds. a) Original: MNIST with its naturally occurring easy and hard examples. b) Partial Observability: MNIST where the bottom two thirds of the images have been cropped out. c) Noisy labeling function: MNIST with shuffled labels for images of 6's, 8's, and 9's. The system seeks a target accuracy $\gamma$ of $98\%$.

## 4.1 VISUALIZING INFERENCE SET DESIGN

To develop intuition for inference set design we first run experiments on the MNIST dataset. The whole MNIST training set is used as the target set from which agents can acquire samples. The MNIST test set is split 50-50 into a validation set used for early stopping and a test set used for measuring model performance on held-out data inaccessible by agents. In many real-world applications, data used for training ML models contains samples that are inherently difficult, only partially observed, or mislabeled. We investigate these three cases. Specifically, we simulate the challenge of partial observability by removing the lower part of the image (Figure 2b) and simulate labeling noise by shuffling labels for selected digits (see Figure 2c).

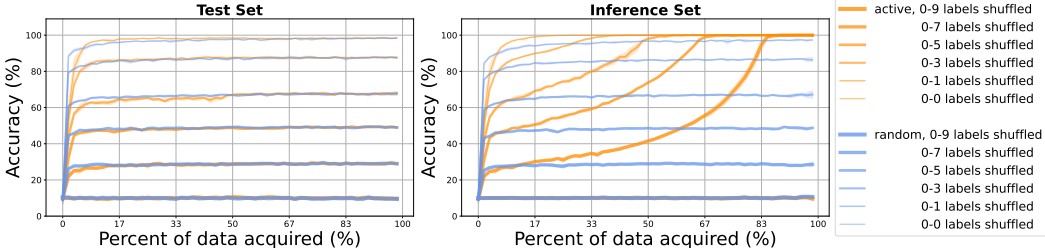

Figure 3: Performance of active and random agents on MNIST subject to an increasing number of shuffled labels. As the task becomes harder, the accuracy gap increases, but the stopping time $\tau$ is triggered later.

In these experiments we study the performance of an active agent using the least-confidence (LC) acquisition function of Equation 3 and a random agent, selecting samples with uniform probability over the inference set. At every active learning step an agent acquires a batch of $1000$ samples from the inference set, adds them to the training set, and retrains the model from scratch. Across these active learning steps, we monitor the accuracy on the inference set, the test set, the acquired batch, and the system accuracy as a whole (see Equation 1). We also record which samples have been acquired at every acquisition step. In all three studied settings in Figure 2, we observe the same phenomenon: the generalisation accuracy measured on the held-out test set quickly saturates for both agents, but the active agent is able to obtain significant improvements on the inference set, and these gains are also reflected in individual acquisition batches and on the system's overall performance.

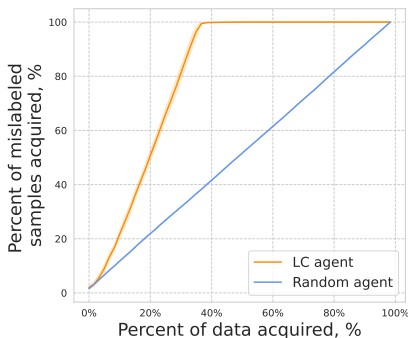

Figure 4: Percent of samples with shuffled labels acquired by each agent throughout the acquisition steps.

The mechanism at play here stems from the active design of the inference set by the agent. We illustrate this for the case of a noisy labeling function in Figure 4. By selectively acquiring the images it tends to mislabel, the active agent makes predicting the labels on the remaining samples in the inference set much easier. Since the majority of samples acquired by the active agent in the first 20 steps represent these challenging examples, its accuracy on the acquired batch is significantly lower than a random agent's (Figure 2c, $3^{\text{rd}}$ row). However, once the process of removing mislabeled samples from the inference set is complete, the active agent's accuracy on the acquired batch drastically increases and surpasses the random agent. To further explore the tradeoff between task difficulty and efficiency gains, we run experiments with shuffled labels for increasingly larger subsets of digits (see Figure 3). The results show that with a higher number of such "noisy" samples, the gap between the active and random agents on the inference set increases. The active agent is able to achieve 100% accuracy on the inference set even with shuffled labels for 80% of the data. However, a more corrupted target set also necessitates more acquisition batches to prune out the difficult examples from the inference set, incurring smaller experimental budget reductions for highly challenging settings.

## 4.2 ASSESSING THE ROBUSTNESS OF THE MECHANISM

In the previous section, we have shown that inference set design leads to significantly improved system performance compared to random sampling, even with highly corrupted datasets. In this section, we validate that these observations hold for a wildly different type of data – molecular datasets. We use the Quantum Machine 9 (QM9) (Ruddigkeit et al., 2012; Ramakrishnan et al., 2014). QM9 contains $134k$ small organic molecules and their quantum chemical properties computed with Density Functional Theory (DFT). For inputs, we convert the SMILES strings molecular representations into their Extended Connectivity Fingerprints (ECFPs). We predict the HOMO-

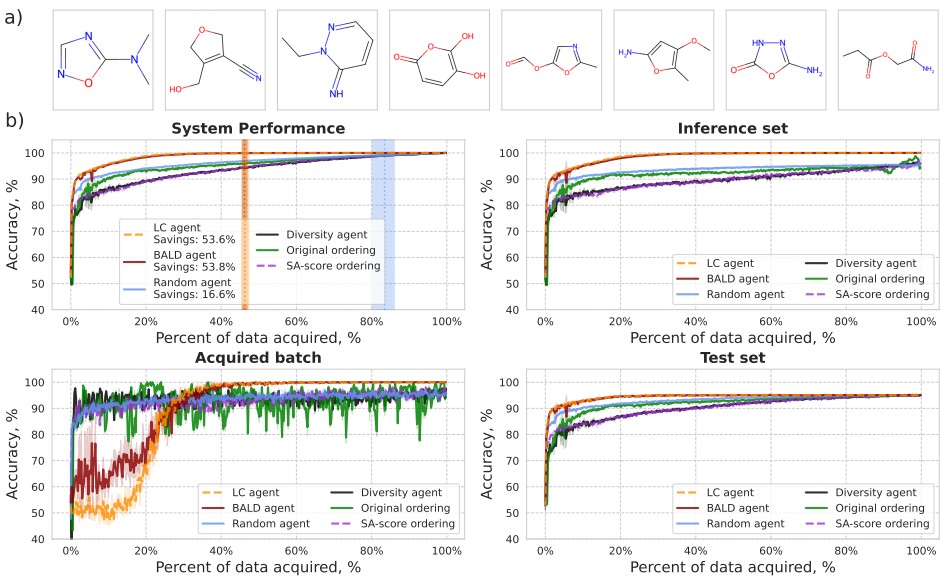

Figure 5: Results on QM9. a) A sample of molecules from the QM9 dataset. b) Performance of active agents (LC and BALD), heuristic orderings, and random sampler on QM9 across 3 seeds.

LUMO gap framed as a classification task by discretizing the values into two balanced classes using median as a boundary condition to explore agents' performance on classification task. Our main active agent uses the least-confidence acquisition function. We also explore the use of the BALD acquisition function (Gal et al., 2017). In addition to active and random agents, for both datasets, we also evaluate the performance of heuristic-based acquisition orderings (original ordering, smallest to largest molecules, ordered by SA-score, etc.) as well as a diversity sampling strategy (selecting molecules with the largest Tanimoto distance to the acquired set).

The results on QM9 demonstrate that both LC and BALD agents achieve much higher accuracy on the inference set than the baselines. Furthermore, they reach near 100% accuracy on both system and inference set after acquiring only 30% of data (see Figure 5). Saturating performance across all agents on the test set suggests once again that the active agent is able to make gains by pruning out difficult examples from the inference set rather than by generalizing better. To confirm this hypothesis, we compiled a list of "hard to predict" molecules by training five models with different random seeds on a 5-fold cross-validation split of the inference set, and flagged the molecules for which at least 4 out 5 models made a mistake. The resulting "hard to predict" subset contains $4,944$ molecules (approximately $5\%$ of the inference set). We then tracked the proportion of these samples that was acquired by each agent in the active learning experiments. Figure 6 shows that the active agents acquire molecules from the "hard to predict" subset at a much higher rate than the baselines.

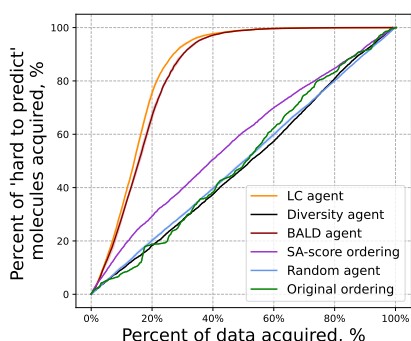

Figure 6: Percent of "hard" examples acquired by each agent throughout the acquisition loop on QM9.

In addition to the classification results on QM9 presented here, we extend our method to a regression task on the much larger Molecules3D dataset in Appendix E.1 and get to a similar conclusion. Put together, these results on additional datasets spanning both classification and regression tasks and including 3 different active acquisition functions confirm and extend the observations made on MNIST in the previous section. They suggest that inference set design is an effective approach to guide acquisition of molecular compounds, by significantly reducing the cost of acquiring a fixed set of molecules and ensuring high system performance. In the next section, we apply this method to real-world biological data.

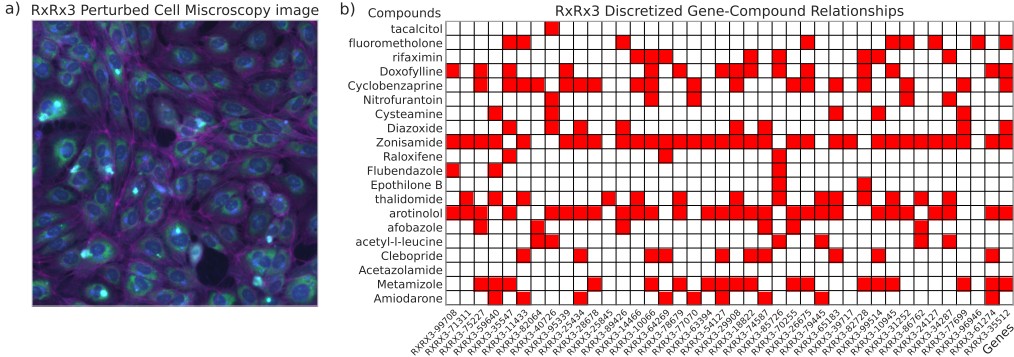

Figure 7: Visualisation of the RxRx3 dataset. a) A microscopy image of perturbed cells. b) A sample of biological map showing whether each pair of gene-based and compound-based perturbations is pheno-similar (in red) or not (in white).

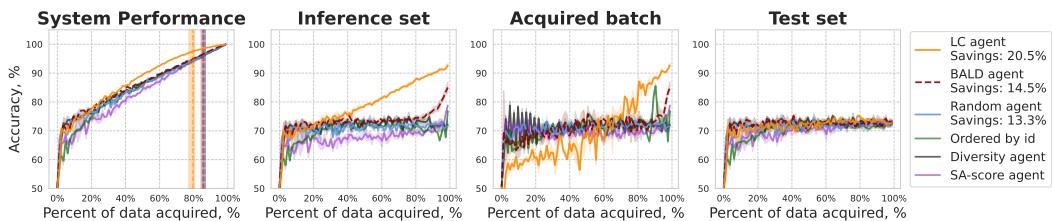

Figure 8: Performance on a pheno-similarity classification task on our proprietary phenomics dataset. The least-confidence agent is shown to be the most effective on this task and reaches the targeted system accuracy of 98% at around 80% of the acquisition procedure.

### 4.3 INFERENCE SET DESIGN IN THE REAL WORLD

Modern HTS platforms combined with genetic perturbation techniques facilitate large scale cell microscopy experiments that are designed to capture the effects of biological perturbations (Celik et al., 2022). In these experiments before taking microscopy images, cells undergo perturbation by either a CRISPR/Cas9-mediated gene knock-out or injection of a bioactive molecular compound at a given concentration. The obtained cell microscopy images are then processed and passed through a neural network to obtain embeddings that correspond to a specific perturbation. With data from such HTS pipeline it is possible to build "Maps of Biology" that contain organized information about known and new biological relationships. Each point on the map corresponds to the cosine similarity between perturbation embeddings and encodes whether two perturbations are related. Expanding maps of biology might uncover new biological relationships that in turn could guide the discovery of leads for new medicines. However, even with ultra-HTS platforms, experimentally acquiring microscopy images of all possible cell perturbations is unfeasible. We aim to apply inference set design to this problem with the goal of reducing costs associated with building maps of biology.

For our experiments, we start by using the publicly available RxRx3 dataset (Fay et al., 2023) which contains the learned embeddings for $17,063$ CRISPR/Cas9-mediated gene knock-out perturbations, as well as $1,674$ FDA-approved compound perturbations at $8$ concentrations each. We frame this as a classification problem and train our models to predict the discretized cosine similarity between the perturbation embeddings of any gene-compound pair. As preprocessing, for each gene-compound pairs, we select the compound-concentration perturbation that yields the largest cosine-similarity with the target gene. The cosine-similarities are discretized by computing a gene-specific threshold using the upper semi-interquartile range (SIQR) with a step-size of $2.5$. Finally, we remove the genes that have an activity rate (across compounds) below $25\%$. The result is a $151 \times 1,674$ matrix of labels encoding either pheno-similarity ($y = 1$) or an absence of relationship ($y = 0$) between a gene and a compound perturbation (see Figure 7b).

For this prediction task, we treat all gene embeddings as known and focus only on the much larger space of compound acquisition. To featurize compounds we use embeddings of the MolGPS model (Sypetkowski et al., 2024). At every active learning step, agents acquire a batch of compounds from the inference set and uncover their relationships with all genes in the dataset (entire row of the matrix). The results on the RxRx3 dataset show only minor improvements over a random agent (see Appendix E). We believe this is due to high complexity of biological relationships and limited size of the inference set. Indeed, successfully learning to infer cellular similarity between gene and compound perturbations from only a few hundred examples is highly unlikely. To push this analysis further, we re-run the same experiment but on our proprietary phenomics dataset. It is similar to RxRx3 but much larger. It contains cell perturbation embeddings for 102,855 compound and 5,580 gene perturbations, leading to a $2,000$ times larger relationship matrix. Using inference set design, the active agent is now able to outperform the baselines on this task, as shown in Figure 8. These experiments represent promising results for large scale applications of active learning to hybrid screens for early drug discovery.

### 4.4 EMPIRICAL VALIDATION OF WEAK CALIBRATION AND STOPPING CRITERION

In this last experimental section, we validate the theoretical claims made in Section 2.2. The proposed stopping criterion assumes weak calibration i.e. that the true accuracy of the model does not need to be exactly equal to its confidence (perfect calibration), but that the correct ordering is conserved. Figure 12 in Appendix F.1 shows that this assumption is empirically verified in all of our experiments, supporting the claim that weak calibration can be achieved in practice. In Figure 13, in Appendix F.2, we also empirically validate that the proposed stopping criterion making use of the bound presented in Equation 5 does indeed cause the agent to stop acquiring new labels at a system accuracy superior to the targeted threshold (see Equation 4).

## 5 DISCUSSION

While active learning has undergone significant theoretical development and experimentation over the past 30 years, its widespread adoption in the industry remains a challenge (Reker, 2019). Active learning is often framed as a means to improve generalization performance on held-out test sets, which is a goal made difficult by real-world challenges such as noisy labeling functions as well as high diversity and partial observability of the input space. However, we believe that a shift in perspective can help unlock the untapped potential of active learning. By focusing on optimizing performance on a target set of samples and allowing the model to decide which examples to label and which to predict, active learning can become a powerful tool for real-world applications.

We applied this perspective to the problem of hybrid screens in drug discovery, where the goal is to acquire readouts for only a subset of compounds while making accurate predictions for the remaining ones. Our approach makes use of a confidence-based acquisition function and leverages the concept of inference set design, a strategy where the model selects the most challenging examples for labeling, leaving easier cases for prediction. Our empirical results, across image and chemical datasets, as well as a real world biological application, show that this approach leads to consistent and significant improvements. Moreover, our results highlight that heuristic-based orderings, often used in real-world data acquisition efforts, are highly suboptimal. They can lead to drastically different results across tasks and datasets and often end up harming the performance by introducing unjustified biases in the acquisition ordering. Importantly, this bias then prevents the experimenter from using the performance on the acquisition batch as an indication of the performance on the remaining examples in the inference set. A confidence-based agent also introduces a bias in sample selection, but assuming weak calibration, this bias is directional and still allows the acquisition batch to be used as a lower bound for the performance on the remaining examples (see Lemma 1). It then compensates for the conservativeness of its stopping criterion by designing its own inference set, leading to overall earlier stopping time and greater budget reductions.

An important limitation of our work is the assumption that observed labels are deterministic and constitute perfect accuracy. Many real-world biological experiments however produce noisy observations. Combining the inference set design framework with aleatoric uncertainty modeling to better address cases of weak supervision represents a promising direction for future work.

ACKNOWLEDGMENTS

We are grateful to the entire team at Recursion and Valence Labs for participating in insightful discussions and providing feedback on this work. We also wish to specifically thank Austin Tripp for offering valuable input on this work as well as Oscar Mendez-Lucio and Peter McLean for their precious support with data processing and for engaging in fruitful exchanges about predictive models for phenomics.

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

# A    THEORETICAL SUPPLEMENTS

## A.1    PROOF OF LEMMA 1

For completeness, we start by re-stating Lemma 1 in more detail.

We consider a multi-class classification problem, with $\mathcal{K} := \{1, 2, \ldots, K\}$ the set of $K$ labels and $\hat{p}(x) := (\hat{p}_1(x), \ldots, \hat{p}_K(x))$ the predicted probabilities for $x$ that $y = k$ for all $k \in \mathcal{K}$. We denote $y_i$ the true label for a sample $x_i$, $\hat{y}_i := \arg\max_k \hat{p}_k(x_i)$ the predicted class by the model and $\hat{v}_i := \max_{k \in \mathcal{K}} \hat{p}_k(x_i)$ the predicted probability value for that class. In the context of sequential label acquisition, we denote $\mathcal{X}_{\text{inf}}^{t-1} := \{x_i\}_{i=1}^N$ the inference set containing $N$ samples at acquisition step $t - 1$, and $\mathcal{D}_{\text{inf}}^{t-1} := \{(x_i, y_i)\}_{i=1}^N$ the same set with the associated labels $y_i$. Stepping forward, at time $t$, an acquisition batch of $N_b$ samples and their labels $\mathcal{D}_b^t := \{(x_i, y_i)\}_{i=1}^{N_b}$ are selected from the inference dataset of the previous step $\mathcal{D}_{\text{inf}}^{t-1}$.

Following the least-confidence acquisition function, at time-step $t$, the samples in $\mathcal{D}_{\text{inf}}^{t-1}$ are ordered by confidence where $x_1$ has the predicted class with the lowest confidence and $x_N$ has the predicted class with the highest confidence:

$$\underbrace{\hat{v}_1 \leq \hat{v}_2 \leq \cdots \leq \hat{v}_{N_b}}_{\text{first } N_b \text{ samples}} \leq \underbrace{\hat{v}_{N_b+1} \leq \cdots \leq \hat{v}_N}_{\text{last } N-N_b \text{ samples}} \tag{6}$$

and the selected batch consists of the first $N_b$ samples. The remaining inference set is composed of the remaining $N - N_b$ samples, i.e. $\mathcal{D}_{\text{inf}}^t := \mathcal{D}_{\text{inf}}^{t-1} \setminus \mathcal{D}_b^t = \{(x_i, y_i)\}_{i=N_b+1}^N$.

The practical utility of this ordering depends on the calibration of the model. Calibration reflects the correctness of the model's confidence compared to its true performance. A perfectly calibrated model $\hat{p}$ would output confidence levels that perfectly match the true probability of correct classification. We extend the notation of Guo et al. (2017) to be dataset-specific, and denote $P_{\mathcal{D}}[Y = \hat{Y}]$ the probability of $\hat{p}$ producing the correct class prediction for any sample from a dataset $\mathcal{D}$, which can equivalently be thought of as $\hat{p}$'s accuracy on that dataset:

$$P_{\mathcal{D}}[Y = \hat{Y}] := E_{(x_i,y_i)\sim\mathcal{D}}[\mathbf{1}(y_i = \hat{y}_i)] \tag{7}$$

Similarly, $P_{\mathcal{D}}[Y = \hat{Y}|\tilde{v}]$ denotes the *confidence-conditional* accuracy of the model for $\tilde{v}$, a specific confidence level[3]:

$$P_{\mathcal{D}}[Y = \hat{Y}|\tilde{v}] := E_{(x_i,y_i)\sim\mathcal{D}}[\mathbf{1}(y_i = \hat{y}_i)|\hat{v}_i = \tilde{v}] \tag{8}$$

Perfect calibration on the inference set would thus imply that for any confidence level $\tilde{v} \in [0, 1]$, we have $P_{\mathcal{D}_{\text{inf}}^{t-1}}[Y = \hat{Y}|\tilde{v}] = \tilde{v}$. Instead, in Lemma 1, we only assume *weak calibration* of the model on the predicted class, i.e. that the confidence *ordering* of the model's top predictions reflects the ordering of probabilities of correct classification. Formally, weak calibration on $\mathcal{D}_{\text{inf}}^{t-1}$ implies that for any pair of samples $x_i$ and $x_j$ in our dataset $\mathcal{D}_{\text{inf}}^{t-1}$, we have:

$$\hat{v}_i \leq \hat{v}_j \implies P_{\mathcal{D}_{\text{inf}}^{t-1}}[Y = \hat{Y}|\hat{v}_i] \leq P_{\mathcal{D}_{\text{inf}}^{t-1}}[Y = \hat{Y}|\hat{v}_j] \tag{9}$$

We aim to prove that, assuming weak calibration on the inference set, the expected accuracy on the least-confidence batch is bounded by the expected accuracy over the remaining inference set:

$$P_{\mathcal{D}_b^t}[Y = \hat{Y}] \leq P_{\mathcal{D}_{\text{inf}}^t}[Y = \hat{Y}] \tag{10}$$

*Proof.* The expected accuracy from Equation 7 can be rewritten in terms of its confidence-conditional form in Equation 8:

$$P_{\mathcal{D}}[Y = \hat{Y}] = \frac{1}{|\mathcal{D}|} \sum_{i=1}^{|\mathcal{D}|} P_{\mathcal{D}}[Y = \hat{Y}|\hat{v}_i] \tag{11}$$

From the weak calibration assumption in Equation 9, it follows that the model's confidence-conditional accuracy is a non-decreasing function of its confidence. Combined with the acquisition

---

[3]In practice such confidence levels would be binned together such that $P_{\mathcal{D}}[Y = \hat{Y}|\tilde{v}]$ represents the accuracy over all samples in $\mathcal{D}$ for which the model's confidence is in $[\tilde{v} - \epsilon, \tilde{v} + \epsilon]$, see (Guo et al., 2017).

mechanism of Equation 6, we obtain an ordering of the probability that the model obtains the correct label along the entire inference set at step $t - 1$:

$$\underbrace{P_{\mathcal{D}_{\text{inf}}^{t-1}}[Y = \hat{Y}|\hat{v}_1] \leq \cdots \leq P_{\mathcal{D}_{\text{inf}}^{t-1}}[Y = \hat{Y}|\hat{v}_{N_b}]}_{N_b \text{ first terms}} \leq \underbrace{P_{\mathcal{D}_{\text{inf}}^{t-1}}[Y = \hat{Y}|\hat{v}_{N_b+1}] \leq \cdots \leq P_{\mathcal{D}_{\text{inf}}^{t-1}}[Y = \hat{Y}|\hat{v}_N]}_{N - N_b \text{ remaining terms}}$$

$$(12)$$

The empirical average of a set of numbers that are inferior or equal to those of a second set has to be inferior or equal to the empirical average of the second set:

$$\frac{1}{N_b} \sum_{i=1}^{N_b} P_{\mathcal{D}_{\text{inf}}^{t-1}}[Y = \hat{Y}|\hat{v}_i] \leq \frac{1}{N - N_b} \sum_{i=N_b+1}^{N} P_{\mathcal{D}_{\text{inf}}^{t-1}}[Y = \hat{Y}|\hat{v}_i] \tag{13}$$

By definition of $\mathcal{D}_{\text{inf}}^t := \mathcal{D}_{\text{inf}}^{t-1} \setminus \mathcal{D}_b^t$, the left-hand side from Equation 13 thus captures all the terms from $\mathcal{D}_b^t$ and the right-hand side those from $\mathcal{D}_{\text{inf}}^t$. Rewriting each side using Equation 11, we obtain:

$$P_{\mathcal{D}_b^t}[Y = \hat{Y}] \leq P_{\mathcal{D}_{\text{inf}}^t}[Y = \hat{Y}] \tag{14}$$

This concludes the proof. $\qquad\square$

## A.2 BOUND DERIVATION

Here we derive the bound presented in Equation 5. The Chernoff bound (Chernoff, 1952) for Bernouilli random variables provides an exponential tail bound for the true mean $\mu$ of a sequence of Bernouilli distributed random variables $Z_1, Z_2, \ldots, Z_n$. It can be expressed using the Kullback-Leibler Divergence (KL) for Bernouilli distributions (Lattimore & Szepesvári, 2020, page 135, Corrollary 10.4):

$$P(\mu \leq a) \leq \exp\left(-n \cdot \text{KL}(\hat{\mu}||a)\right) \quad \forall \quad a \in [0, \hat{\mu}] \tag{15}$$

$$\text{with} \quad \text{KL}(\hat{\mu}||a) := \hat{\mu} \log \frac{\hat{\mu}}{a} + (1 - \hat{\mu}) \log \frac{1 - \hat{\mu}}{1 - a} \tag{16}$$

It quantifies the probability that the true mean $\mu$ is smaller or equal to some bound $a$ given the observed sample mean $\hat{\mu}$ and sample size $n$. For us, the value $\{0, 1\}$ of a variable $Z_i$ indicates whether a particular sample $x_i$ was correctly classified i.e. $Z_i := \mathbf{1}(y_i = \hat{y}_i)$. In the context of inference set design and following the notation from Section 2, $\hat{\mu}_b^t$ represents the observed accuracy on the acquired batch at time-step $t$, with acquisition batch-size $N_b$, and $\mu_{\text{inf}}^t$ represents the (unknown) accuracy of the model on the remaining inference set. Note that the Chernoff bound usually assumes that $\hat{\mu}_b^t$ is an unbiased estimator of $\mu_{\text{inf}}^t$. In our case, $\hat{\mu}_b^t$ is a biased estimator due to the active selection mechanism of the batch. However, in Lemma 1, we show that this estimator is actually a conservative estimate of the true accuracy, and thus in turn contributes to making the bound presented in Equation 15 even more conservative, and preserves its validity.

To establish a confidence level on that bound, we can lower-bound the right-hand side itself to the desired bound-failure probability $\delta$, which after rearranging yields:

$$\text{KL}(\hat{\mu}||a) \geq \frac{\log(\frac{1}{\delta})}{N_b} \tag{17}$$

At time $t$, we thus seek the maximum bound value $a = \alpha_t$ for $\mu_{\text{inf}}^t$ such that the inequality on Equation 17 holds by finding the value of $a$ that satisfies the following condition:

$$\alpha_t = \min_a \left\{ a \in [0, \hat{\mu}_b^t] : \text{KL}(\hat{\mu}||a) \leq \frac{\log(\frac{1}{\delta})}{N_b} \right\} \tag{18}$$

Since this is a scalar optimization problem from closed-form expressions, computing $\alpha_t$ can be done easily and efficiently using a grid-search. With this choice for $a$, we obtain the desired probabilistic bound $P(\mu_{\text{inf}}^t < \alpha_t) \leq \delta$ summarized in Equation 5.

## B  ALGORITHM

---

**Algorithm 1** Hybrid Screen using Inference Set Design

---

1: **Input:** Acquisition batch-size $N_b$, threshold $\gamma$, margin $\delta$
2: **Initialize** Step t=0, observation set $\mathcal{X}_{\text{obs}}^{t=0} \leftarrow \emptyset$, inference set $\mathcal{X}_{\text{inf}}^{t=0} \leftarrow \mathcal{X}_{\text{target}}$, predictor $\hat{p}$.
3: **repeat**
4:     Train the predictor $\hat{p}$ on $(\mathcal{X}, \mathcal{Y})_{\text{obs}}^t$ (if not empty)
5:     Obtain the predictions on the inference set $\hat{\mathcal{P}}_{\text{inf}}^t \leftarrow \{\hat{p}(x_i) \; \forall x_i \in \mathcal{X}_{\text{inf}}\}$
6:     Run acquisition function to obtain scores $\mathcal{S}_{\text{inf}}^t \leftarrow g(\hat{\mathcal{P}}_{\text{inf}}^t)$
7:     Select a batch of $N_b$ inputs with the highest scores $\mathcal{S}_{\text{inf}}^t$ to form $\mathcal{X}_b^t$
8:     Remove the acquired batch from the inference set $\mathcal{X}_{\text{inf}}^t \leftarrow \mathcal{X}_{\text{inf}}^{t-1} \setminus \mathcal{X}_b^t$
9:     Obtain the true labels $\mathcal{Y}_b^t$ for the acquisition batch
10:     Append the acquired batch to the observation set $(\mathcal{X}, \mathcal{Y})_{\text{obs}}^t \leftarrow (\mathcal{X}, \mathcal{Y})_{\text{obs}}^{t-1} \cup (\mathcal{X}, \mathcal{Y})_b^t$
11:     Compute $\alpha$ on $(\mathcal{X}, \mathcal{Y})_b^t$ from Equation 5
12: **until**
$$\frac{|\mathcal{X}_{\text{obs}}^t| + \alpha|\mathcal{X}_{\text{inf}}^t|}{|\mathcal{X}_{\text{target}}|} > \gamma \quad \text{or} \quad \mathcal{X}_{\text{inf}}^t = \emptyset$$

13: **Return** hybrid screen readouts: $(\mathcal{X}, \mathcal{Y})_{\text{obs}}^{t=\tau} \cup (\mathcal{X}, \hat{\mathcal{Y}})_{\text{inf}}^{t=\tau}$

---

## C  ADDITIONAL DETAILS ON DATASETS AND PREPROCESSING

### C.1  MOLECULAR DATASET PREPROCESSING

In many practical applications, exact geometries of screened molecules are unknown as they require computationally expensive DFT calculations. As a first data processing step, we use the RDKit (Landrum et al., 2024) and Molfeat (Noutahi et al., 2023) libraries to convert molecular structures into SMILES strings and compute their Extended Connectivity Fingerprints (ECFPs). Both molecular datasets are cleaned by removing duplicated SMILES and fingerprints as well as single-atom structures. For total energies in the Molecules3D dataset we use a reference correction technique where atomic energies are calculated using a linear model fitted to the counts of atoms of each type present in a molecule (the obtained atomic energies are presented in Table 1). For reference correction, a randomly selected sample of 100k molecules is used. The atomic energies are then subtracted from the total energies of all molecules in the dataset. The obtained referenced-corrected energies are normally distributed with mean around 0 eV. A small number of outliers with reference-corrected energy values above 10 standard deviations are removed from the dataset as well as the 100k samples that were used for reference correction to avoid data leakage.

The final QM9 and Molecule3D datasets contain $133,885$ and $3,453,538$ molecules respectively. Both datasets are split into inference, validation, and test sets with 80%, 5%, 15% fractions. The QM9 HOMO-LUMO gap values are discretized into 2 balanced classes using median as a boundary condition to explore agents' performance on a classification task.

| Atomic Number | Energy (eV) | Atomic Number | Energy (eV) |
|:---:|:---:|:---:|:---:|
| 1 | -26.765 | 14 | -7922.180 |
| 5 | -673.550 | 15 | -9313.610 |
| 6 | -1054.411 | 16 | -10810.329 |
| 7 | -1483.001 | 17 | -12474.680 |
| 8 | -2034.056 | 32 | -56538.647 |
| 9 | -2687.455 | 33 | -60828.588 |
| 12 | -5367.759 | 34 | -65296.349 |
| 13 | -6657.476 | 35 | -69980.303 |

Table 1: Atomic energies used for reference correction on Molecules3D dataset.

## C.2 RxRx3 DATASET PREPROCESSING

The RxRx3 dataset contains one embedding for each well. Each perturbation type (gene-guide pair or compound-concentration pair) has several replicates across wells, plates and experiments. Each plate also contains unperturbed control cells which are used to keep track of and eliminate a portion of the batch effects (Sypetkowski et al., 2023). These raw embeddings thus need to be aligned and aggregated. We align them by centering and scaling each perturbation embedding to the embeddings of the experiment-level unperturbed control wells. The embeddings are then aggregated through a multi-stage averaging procedure, across wells, plates, experiments and guides (for CRISPR perturbations), which yields an average embedding for each gene-perturbation and each compound-concentration perturbation. We then use the obtained embeddings to compute the cosine similarities between gene and compound perturbations in the RxRx3 dataset.

## D HYPERPARAMETERS AND IMPLEMENTATION DETAILS

For all experiments presented in this work we use MLP models with residual connections (Touvron et al., 2021). All experiments were repeated with 3 different random seeds and the hyperparameters are summarized in Table 2.

| **Hyperparemeter name** | MNIST | QM9 | Molecules3D | RxRx3 | Proprietary data |
|---|---|---|---|---|---|
| Acquisition batch size | 1,000 | 250 | 10,000 | 10 | 1000 |
| Number of hidden layers | 2 | 3 | 2 | 2 | 2 |
| Hidden layer size | 512 | 512 | 512 | 512 | 1024 |
| Learning rate | 0.001 | 0.001 | 0.001 | 0.001 | 0.001 |
| gradient norm clip | 1.0 | 1.0 | 1.0 | 1.0 | 1.0 |
| Dropout | 0.1 | 0.1 | 0.1 | 0.1 | 0.1 |
| Train epochs | 1,000 | 1,000 | 30 | 30 | 1000 |
| Train batch size | 1,024 | 1,024 | 32,768 | 1,024 | 1,024 |
| Early stop patience | 50 | 50 | 15 | 25 | 25 |
| Number of ensemble members | - | - | 5 | - | - |

Table 2: Hyperparameters for experiments.

# E   ADDITIONAL RESULTS

## E.1   REGRESSION TASK ON MOLECULES3D

To evaluate the inference set design paradigm on a regression task we use the Molecules3D dataset (Xu et al., 2021). Molecules3D contains structures and DFT-computed properties of nearly 4 million molecules. In our experiments we aim to predict the HOMO-LUMO gap and total energy. For inputs, we convert the SMILES strings molecular representations into their Extended Connectivity Fingerprints (ECFPs) (see Appendix C.1 for all details).

For this regression task, we use a query-by-committee (QBC) active learning approach that computes variance across the predictions of an ensemble. To determine the stopping time we use a criterion with two parameters: MSE threshold $t_{MSE}$ on the acquired batch and patience $p$. The stopping time is reached if the acquired batch MSE is lower than $t_{MSE}$ for $p$ steps. Like for our QM9 experiments, in addition to active and random agents, we also evaluate the performance of heuristic-based acquisition orderings (molecules ordered by size, sorted by SA-score, etc.). QBC achieves an approximately five times lower MSE compared to the random agent or heuristic-based orderings (see Figure 9). This shows that inference set design approach is not limited to classification tasks and can be applied to regression problems.

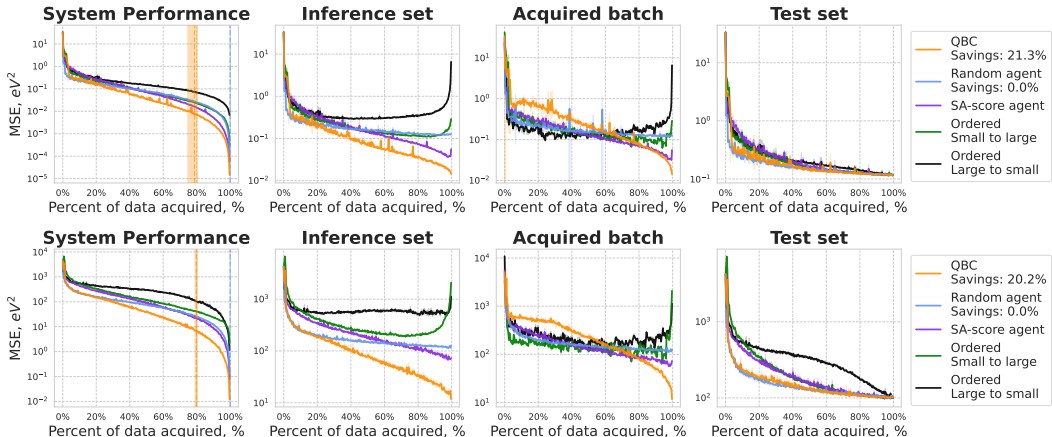

Figure 9: Performance of QBC, random, and heuristic agents for HOMO-LUMO gap prediction (top) and energy prediction (bottom) on Molecules3D dataset. The dashed vertical shows average stopping time across random seeds, surrounded by mix to max intervals (left to right). The patience parameter $p = 10$ was used in all experiments with the threshold MSE $t_{MSE} = 0.1eV^2$ for HOMO-LUMO gap and $t_{MSE} = 100eV^2$ for Energy predictions. QBC agent satisfied stopping condition after acquiring $\approx 80\%$ of the data.

Although, in many applications predicting properties of larger molecules presents a more challenging task, our experiments on the Molecules3D dataset demonstrate that acquiring molecules ordered from large to small may harm the predictions on inference set and overall system performance (see Figure 9). One of the reasons is the distribution of chemical elements across molecules in the dataset. When acquiring molecules from small to large, all unique chemical elements of the Molecules3D dataset are present in the training set after acquiring just the first 1,000 samples. However, when acquiring molecules from large to small, some chemical elements remain only in the inference set until the very end of the experiment which is especially detrimental to the total energy predictions (see Figure 10). This result demonstrates that using heuristic rules such as ordering molecules by size for data acquisition does not guarantee optimal acquisition or generalization of heuristic rules to new data and instead risks harming the performance due to the introduction of such biases.

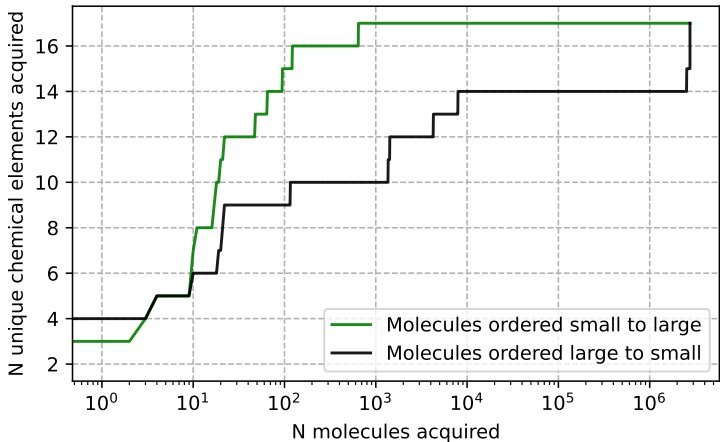

Figure 10: Number of unique chemical elements in the training set when acquiring samples ordered by molecule size. Acquiring molecules from large to small leaves several chemical elements out of the training set until the end of the dataset.

## E.2 CLASSIFICATION TASK ON RxRx3

Our results on the RxRx3 dataset demonstrate minor improvements from the active agents (LC and BALD) compared to random selection. The accuracy on the inference and test sets remains low throughout the experiment regardless of the acquisition method. This is unsurprising considering the extreme difficulty of predicting biological relationships in a low data regime. The result suggests that in the setting with low predictive power, even when the majority of the data is acquired, the ability of inference set design to provide significant budget reductions is limited.

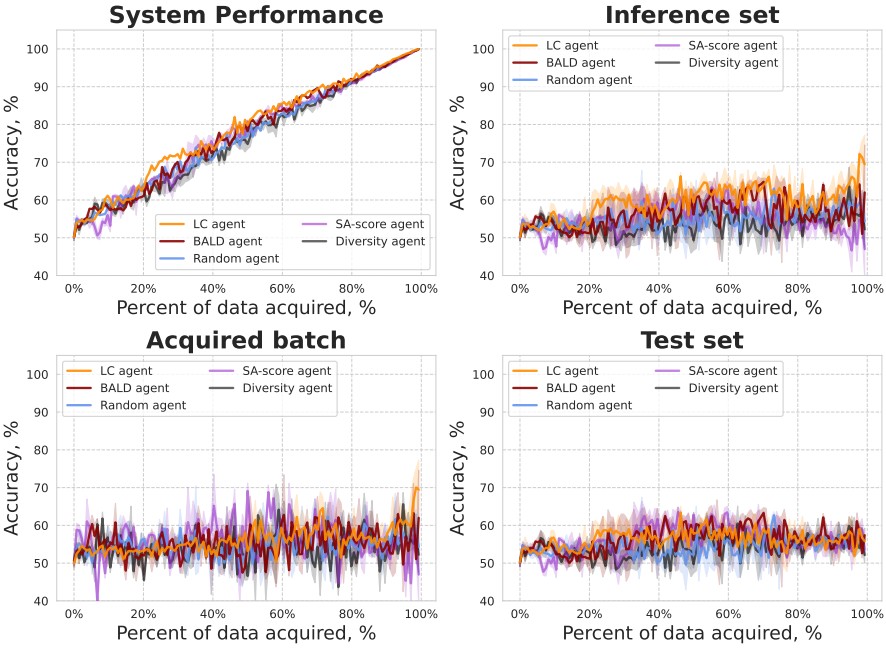

Figure 11: Performance of agents on pheno-similarity classification task on RxRx3.

# F    ADDITIONAL ANALYSIS

## F.1    EMPIRICAL VALIDATION OF WEAK CALIBRATION ASSUMPTION

In Lemma 1, we show that when using a least-confidence acquisition function, at any time-step $t$, the measured accuracy $\hat{\mu}_b^t$ on the acquisition batch is a lower-bound on the unobserved accuracy on the remaining inference set $\hat{\mu}_{\text{inf}}^t$, assuming that the model $\hat{p}$ is weakly calibrated in the inference set. A weakly calibrated model is such that an increased confidence translates to a higher likelihood of correct prediction (higher accuracy). In Figure 12, we can see that at several points throughout the active learning loop, the least-confidence based agent is not *perfectly* calibrated. Indeed, there is a substantial gap between its confidence levels and the true accuracy (seen when comparing the colored bars to the identity function shown in gray). However, the model is generally *weakly* calibrated (the colored bars are always increasing). These observations support the empirical validity of our assumption for Lemma 1.

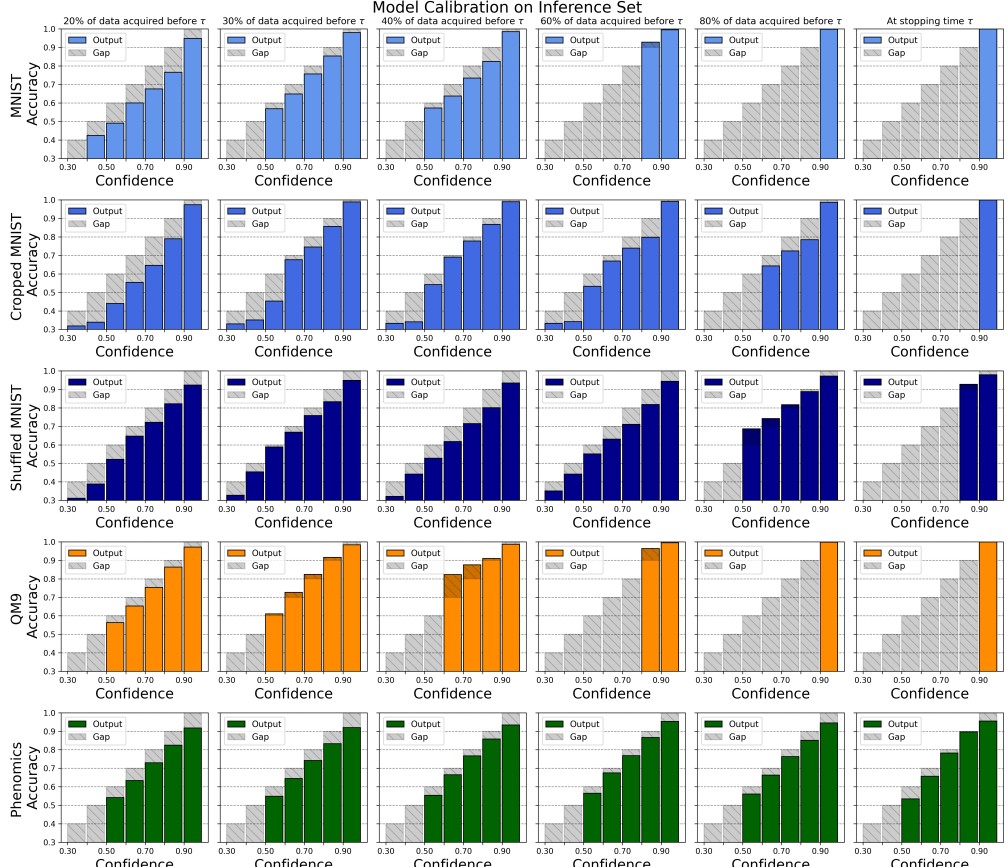

Figure 12: Model calibration analysis for a run of the least confidence agent at different active learning steps (columns) for all of our classification experiments (rows). The colored bars represent the accuracy of predictions binned w.r.t their confidence level, and the gray bars show the identity function illustrating what perfect calibration would look like. The models satisfy the condition of being weakly calibrated since the accuracy of the model's predictions increases monotonically with their confidence. The confidence distribution shifting to the right as $t$ increases indicates the growing confidence of the model in correctly predicting the labels of the remaining examples in the inference set.

## F.2 EMPIRICAL VALIDATION OF STOPPING CRITERION

In Section 2.2, we present a stopping criterion based on the fact that the accuracy measured on the acquisition batch can be used as a proxy for the accuracy on the remaining, non-acquired samples (inference set). The criterion is simple, once the estimated system accuracy $\hat{\mu}_{\text{sys}}^t$ at time $t$ (see Equation 4) is above a user-defined targeted threshold $\gamma$, the acquisition is stopped. This criterion makes use of the bound of Equation 5, and both a random sampler and a least-confidence based agent can use this bound in a principled way. The random agent can validly use it because the accuracy of its acquisition batches, uniformly drawn from the inference set, are unbiased estimates of the true accuracy on the entire inference set, which is typically required for such bounds. The least-confidence agent can use it validly because we show in Lemma 1 that, assuming that the model is weakly calibrated (which we empirically validate in Appendix F.1), because of its confidence-based acquisition function, the accuracy of its acquisition batches represent a *lower-bound* on the true inference set accuracy, making the bound of Equation 5 even more conservative.

To empirically validate this bound (Equation 5), we run a large number of trials for both the least-confidence (LC) and random agents across the classification datasets used in the experiments section. The results show that for the LC agent, all trials ended at a stopping time $t = \tau$ at which the true system accuracy $\mu_{\text{sys}}^t$ was above the threshold $\gamma$, which is in accordance with the fact that the LC agent uses a lower-bound estimate in place of an unbiased estimate for the inference set accuracy, resulting in a looser bound for Equation 5 and an actual failure-probability lower than $\delta$. For the random agent, we observed $1\%$ of the trials end with a system accuracy below the threshold in one of the experiments, $6\%$ in another, and $0\%$ on the three remaining datasets. These results are in accordance with the theory presented in Section 2.2 and Appendix A.2. On QM9, 6 out of 100 experiments resulting in slight bound failure is statistically in accordance with the theoretical bound failure probability of $\delta = 5\%$. For the other datasets, the bound was looser, which is also a possibility, and showcases that the observed gap between $\delta$ and the true bound-failure probability can be problem-dependent.

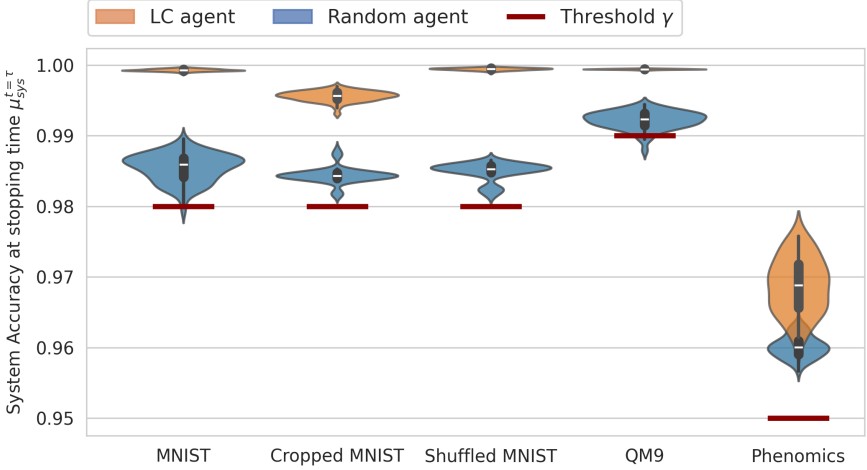

Figure 13: System accuracy at stopping time observed across 100 seeds for the MNIST, Cropped MNIST, Shuffled MNIST and QM9 experiments, and 50 seeds for the Phenomics experiments. In all experiments, we use a bound-failure probability of $\delta = 0.05$. For the LC agent, all trials lead to a stopping time $t = \tau$ at which the true system accuracy $\mu_{\text{sys}}^{t=\tau}$ higher than the threshold $\gamma$. For the Random Agent, we observed the same thing for Cropped MNIST, Shuffled MNIST and Phenomics. We also observed $1\%$ of bound failure for (regular) MNIST, and $6\%$ for QM9.

