# OpenReview forum: "Efficient Biological Data Acquisition through Inference Set Design"
_ICLR.cc/2025/Conference — ICLR 2025 Poster_

### Official Review · Reviewer_Nxf3 · 2024-10-23

**Soundness:** 1
**Presentation:** 1
**Contribution:** 2
**Rating:** 8
**Confidence:** 4

**Summary:**

The paper introduces and tries to tackle the problem of identifying a small subset of data on which to run costly measurements to train a machine learning model which infers the most probable measurements for the remainder of the data, with the theoretical guarantee that the model makes less of $\gamma$ percent of errors at inference time. The paper introduces a stopping criterion that would satisfy the correctness guarantee and applies their corresponding algorithm on several benchmark and real-life data sets.

**Strengths:**

- Originality: The submission tackles an interesting problem, where the goal is not to maximize model generalization but to ensure model correctness on a specific subset of data. This is a first step towards better theoretically-backed approaches in biology (compared for instance to what was done in the LINCS L1000 data set [1] for the regression of gene expression, where they only selected a fixed subset of genes to measure based on prior knowledge instead of difficulty in predicting gene expression).

- Quality: The experimental results look good, with appropriate baselines (random selection of new samples to measure, or adding a next batch to measure given the original order of samples). There is a good effort in applying the method to benchmark and real-life, biological data.

[1] Subramanian, Aravind, et al. "A next generation connectivity map: L1000 platform and the first 1,000,000 profiles." Cell 171.6 (2017): 1437-1452.

**Weaknesses:**

- Clarity: **(Major concern)** The full paper is not very well-written:

(1) The definition of $b$ (or $N_b$ in the pseudocode in appendix) is unclear and never defined. In Equation (5), it seems that $b=|X^t_\text{obs}|$ that is, the total number of points measured up at time $t$.

In Lemma 1, $b=|X^t_b|$, which is the total number of new points added to the observed set $X^{t-1}_{obs}$ at time $t$.

$N_b=|X^t_b|$ in the pseudocode in appendix.

In the proof of Lemma 1 in appendix, $b=|X^t_{inf}|$, that is, the number of points that are left to infer at time $t$ (there is no reason to assume that $b=|X^t_{inf}|=|X^t_b|$...).

Moreover, it should be mentioned earlier than the appendix or the experimental section that a **constant** number of samples is added to the measured point set.

(2) Perhaps a good thing would be to define all variables ($X^t_\text{obs}$, $X^t_b$, etc.) in a single paragraph just after the introduction. Please don’t remove the time-dependent indices ($X_\text{obs}$, $\hat{f}$…) and the hats ($f$ instead of $\hat{f}$ in the proof of Lemma 1), as it makes everything very confusing.

(3) Why mention the query-by-committee sampling, as there is no theoretical guarantee associated with it?

(4) It seems that $\hat{f}$ has a range in $[0,1]$ (or $\mathbb{N}$) whereas $\hat{p}$ has a range in $[0,1]$. Then writing $\hat{p}=\hat{f}(X)$ in the proof of Lemma 1 is extremely confusing.

(5) **(Minor suggestion)** I would put the related works section earlier in the manuscript, for instance, when the differences between active learning and inference set design is discussed, as I think it would make more sense.

(6) **(Minor suggestion)** A one-liner of the full structure of the Algorithm should be present at the end of the section 2 with a link to the pseudo-code in the appendix, so that the algorithmic contribution is clearly mentioned.

- Quality/Significance: **(Major concern)** As it is, the proof of Lemma 1 is wrong. What is $\hat{Y}$ ? the difference with $Y$? Is one the prediction made by the model $\hat{f}$ and the other the true measured label (and then $\hat{Y}= \hat{f}(X)$)? However, $y$ seems to be the probability given by the model in Equations (3) and (4). The proof displays the opposite sign of what is stated in Lemma 1 in the main text. Please rewrite properly the proof, including the use of Bayes’ theorem (?) in the second equality, the problem of the definition of the variables $b$, $\hat{f}$, $\hat{Y}$, $\hat{p}$, why “$P(\hat{p}=\hat{f}(X))$”=1 (assuming that here $\hat{f}(X)$ is the probability returned by the model: is it related to the model being accurate on every measured point?), and how the induction works (at least) from T-1 to T-2.

The result shown in Lemma 1 is not empirically tested (that is, at the stopping point of the Algorithm, with a prespecified probability $1-\delta$, the error is less than $\gamma$). The code producing the experiments is not shared, as such, the research is not reproducible.

- Significance: **(Minor concern)** The choice of accuracy measure is perhaps not very appropriate for the data: for instance, we expect more drugs to be labeled negatively (for instance, in drug discovery) than positively. Area Under the Curve is said to be slightly more appropriate on imbalanced data [2].

[2] Ling, Charles X., Jin Huang, and Harry Zhang. "AUC: a better measure than accuracy in comparing learning algorithms." Advances in Artificial Intelligence: 16th Conference of the Canadian Society for Computational Studies of Intelligence, AI 2003, Halifax, Canada, June 11–13, 2003, Proceedings 16. Springer Berlin Heidelberg, 2003.

The proof of Lemma 1 being incorrect/badly-written is why I rated soundness to 1. The definitions of notation not being clear to the point of impairing the understanding of the paper is why I rated presentation to 1. The interesting question (mitigated by the perhaps incorrect Lemma 1) is why I rated contribution to 2. The main reason why I rated the paper 1 is because of the proof of Lemma 1.

**Questions:**

Typos:
- Is “weakly calibrated” (page 2) the same as “approximately calibrated” (Lemma 1)?
-  In Lemma 1, $P[\hat{Y} = Y \mid \hat{p} = p_1 ] > P [\hat{Y} = Y \mid \hat{p} = \underline{p_2} ]$ for $p_1 > p_2$
- In Equation (5), please mention that KL is the Kullback-Leibler divergence… also mention the name of the result (Corollary 10.4) instead of the page.

Questions:
- What does “treat performance on $X^t_\text{inf}$ as though they were IID samples from $X^t_\text{inf}$” (Line 213) mean? Independence of the probabilities given by the model across data samples from $X^t_\text{inf}$?
- **(Score-changing question)** Please provide a proof for Lemma 1 that solves my concerns listed above.
- **(Score-changing question)** Is there a closed-form expression or at least an upper bound on the stopping time $\tau$ in Equation (6) (as the whole selection process is deterministic as long as the prediction model is deterministic)? If so, is the subsequent budget of measurements (that is, $|X^t_\text{obs}|$ where t is the final selection step of the algorithm) realistic with respect to the total number of items (for instance, we might want a number of measured elements sublinear in N for a reasonable value of $\gamma$, where N is the total number of items in the data set)?
- **(Minor question)** At each time t, the model $\hat{f}^t$ (or $\hat{f}^t_1, … \hat{f}^t_{Nmodels}$ for the query-by-committee) has to be retrained on the full $X^t_{obs}$ and evaluated on $X^t_{inf}$, and then take the arg max on $X^t_{inf}$ (least confidence sampling and query-by-committee). How expensive is it in practice? The budget of measurements ($|X^t_{obs}|$) is important but perhaps we would like also to minimize the number of selection rounds (that is, $t$) as training/evaluating/sampling might also be expensive. Minimizing $t$ might however be contradictory with the objective of minimizing $|X^t_{obs}|$ though, as if we measure a slight over-approximation of the minimal $X^t_{obs}$ in a single selection step, we optimize for the former objective but not the latter.
- **(Minor suggestion)** Why not a more adaptive approach where all points such that the maximal proba across class labels is no higher than 0.5 (and then the acquisition batch size is no longer a constant)?
- **(Score-changing questions)** (1) There is a shadowed orange part on the plots in Figure 1, are the curves the average across all considered points (System performance: all points in MNIST training set = $|X_\text{target}|$, inference set $|X_\text{target}|-|X^t_\text{obs}|$ -that changes across the x-axis- acquired batch $1,000$, and test set = 50% of full data set) or the average across random seeds (=3 according to the appendix)?

(2) What the “saving” mentioned in Figure 1? The difference between the number of actually measured points and measuring all samples at a given point in time?

(3) What’s the threshold $\gamma$ used in the experiments for the stopping criterion? Can you check that the theoretical guarantee of Lemma 1 is empirically satisfied (by applying some calibration procedure on the trained model, if needed)?

I understand that not all score-changing questions can be answered, especially in the short time for rebuttal. I would be willing to increase the score whenever one of these can be answered (the most important being Lemma 1 and the ones related to the plots) / if the major concerns listed above are addressed.

Summary of the suggested modifications:
- Provide a clear and consistent definition of $b$ (or $N_b$) early in the paper, ideally in a dedicated notation section.
- Ensure that this definition is used consistently throughout the paper, including in equations, lemmas, proofs, and pseudocode.
- Explicitly state earlier in the paper that a constant number of samples is added to the measured point set in each iteration.
- Clearly define all variables used in the proof of Lemma 1, including $\hat{Y}$, $Y$, $\hat{f}$, and $\hat{p}$, and explain their relationships.
- Revise the proof to ensure consistency with the statement in Lemma 1, particularly regarding the inequality sign.
- Provide a step-by-step explanation of the proof, including justifications for each step (e.g., the use of Bayes' theorem).
- Clarify the induction process, especially the step from T-1 to T-2.
- Explain or justify any assumptions made in the proof of Lemma 1, such as "$P(\hat{p}=\hat{f}(X))$"=1.
- Include empirical tests of Lemma 1 in the experiments, specifically verifying that the error is indeed less than $\gamma$ with probability $1-\delta$ at the stopping point.
- Provide access to the code used for the experiments, either through a public repository or as supplementary material.
- Include more details about the experimental plots.

------------------------
**UPDATE**: I raised my score to 8 after reading the updated manuscript and exchanging with the authors.

---

> ### Author Response · Authors · 2024-11-20
> **Author Response to Reviewer #4 (Nxf3) -- (1/3)**
>
> We wish to thank Reviewer-Nxf3 for a very thorough review, for their numerous suggestions for improvement, and for instigating interesting questions and discussions. The reviewer's main concern was with the presentation of our theoretical contributions. There were indeed important clarity issues with Lemma 1 which we are grateful for the reviewer for highlighting these issues in such detail. We believe these issues have now been fully fixed in the updated manuscript. The reviewer also had a number of additional comments and suggestions. To keep our response organized, we will start by following the summary list that the reviewer has greatfully provided at the end of their review and address each of these points. We then complement this list with additional comments and questions from the earlier parts of the review in an effort to provide an complete and exhaustive response to the entire review.
>
> ## Main concerns and suggestions:
>
> > 1. Reviewer-Nxf3: "Provide a clear and consistent definition of $b$ or $N_b$ early in the paper. Ensure that this definition is used consistently throughout the paper, including in equations, lemmas, proofs, and pseudocode."
>
> Indeed, we did mistakenly use both $b$ and $N_b$ to denote the acquisition batch size in the original submission. We have corrected it to only use $N_b$ in the updated manuscript.
>
> > 2. Reviewer-Nxf3: "Explicitly state earlier in the paper that a constant number of samples is added to the measured point set in each iteration."
>
> This has been clarified (see Section 2.1, updated manuscript).
>
> > 3. Reviewer-Nxf3: "Clearly define all variables used in the proof of Lemma 1, including $y$, $\hat{y}$, $\hat{f}$, $\hat{p}$ and explain their relationships. Revise the proof to ensure consistency with the statement in Lemma 1, particularly regarding the inequality sign. Provide a step-by-step explanation of the proof, including justifications for each step. Explain or justify any assumptions made in the proof. [...] The main reason why I rated the paper 1 is because of the proof of Lemma 1. Please provide a proof for Lemma 1 that solves my concerns listed above."
>
> Thank you for pointing this out. We would greatly appreciate if you could revisit the theoretical sections of the paper (Section 2.2 and Appendix A, updated manuscript). We have deeply reworked the notation and corrected the multiple typos that were impeding the reader's comprehension, and believe that the section is now significantly easier to follow. In particular, in Appendix A.1 (updated manuscript) we provide a much more detailed and better structured statement of Lemma 1 and its proof, and in Appendix A.2 (updated manuscript) we provide a more detailed derivation of the bound presented in Equation 6 which constitutes a central component of our stopping criterion.
>
> Please also note that the central claim of Lemma 1 is identical to the original submission: assuming weak calibration of the model, the accuracy on the acquired batch constitutes a lower-bound on the accuracy of the remaining inference set.
>
> > 4. Reviewer-Nxf3: "Clarify the induction process, especially the step from T-1 to T-2."
>
> This was confusing and we have corrected that part of the explanation (see Appendix A1, updated manuscript). No induction process is required: assuming weak calibration of the model, the accuracy on the acquired batch consitutes a lower-bound on the accuracy on the inference set at any step $t$ of the active learning loop . No induction reasoning from the last step is needed because this result does not concern the accumulation of samples in the observation set $X_{\text{obs}}$. For any step $t$, this result only concerns the acquired batch $X_{b}^t$ and the remaining inference set $X_{\text{inf}}^t$ from which this batch has been removed.

---

> ### Author Response · Authors · 2024-11-20
> **Author Response to Reviewer #4 (Nxf3) -- (2/3)**
>
> > 5. Reviewer-Nxf3: "Include empirical tests of Lemma 1 in the experiments, specifically verifying that the error is indeed less than $\gamma$ with a probability $1 - \delta$ at the stopping point."
>
> Thank you for these great suggestions. We have added additional experiments to investigate the empirical correctness of our assumption and bound  which are briefly presented in Section 4.4 of the updated manuscript. Specifically:
> - to your first suggestion: we have added empirical analysis of the correctness of our weak calibration assumption of Lemma 1 (see Section 2.2, updated manuscript). We showcase these calibration plots for our QM9 experiments in the main text (see Figure 9, updated manuscript) and provide the same result for all of our experiments in Appendix (see Figure 14, updated manuscript).
> - to your second suggestion: we also provide an empirical validation that the true system accuracy $\mu_{\text{sys}}^{t=\tau}$ at the stopping time $t=\tau$ is indeed superior or equal to our target threshold $\gamma$ as described in Equation 4 (updated manuscript). The results of this analysis are presented in Appendix (see Figure 13, updated manuscript).
>
> Overall we believe these investigations greatly improve the paper and confirm (1) that the weak calibration assumption is both useful and reasonable -- indeed, we can see that the model is not *perfectly* calibrated, but the calibration *ordering* is correct -- and (2) that the stopping criterion that we propose making use of a lower bound on the inference set accuracy is both theoretically principled and holds well in practice.
>
> > 6. Reviewer-Nxf3: "Provide access to the code used for the experiments, either through a public repository or as supplementary material."
>
> We also believe that experimental reproducibility is essential and as mentioned in the original submission, it has always been our intention to make the code publicly available upon publication. It can however also be useful to reviewers and we thus have made the code annonymously available at this link for the entire duration of the reviewing process: https://anonymous.4open.science/r/inference_set_design-889D
>
> > 7. Reviewer-Nxf3: "Include more details about the experimental plots. [...] Are the curves the average across all considered points or the average across random seeds?"
>
> Thank you for pointing this out. All of our curves are averaged across 3 random seeds. We have added an explanation of the shaded areas for our experimental graphs at the begining of the experiments section (see Section 4, updated manuscript). We have also added min-max intervals (across seeds) for the stopping times on all of our plots.
>
> ## Additional questions and comments:
>
> > 8. Reviewer-Nxf3: "Why mention the query-by-committee sampling, as there is no theoretical guarantee associated with it?"
>
> This was indeed confusing in the original submission. We were mentioning it in the Methods section since this technique is used for our regression-based experiments (see Appendix E.1, updated manuscript), but this might indeed be confusing as our theoretical garantees focus on the use of least-confidence acquisition function only. For more clarity, in Section 2, we now only introduce the least-confidence acquisition function, and introduce additional acquisition functions which are used as baselines in their respective experimental sections only.
>
> > 9. Reviewer-Nxf3: "A one-liner of the full structure of the Algorithm should be present at the end of the section 2 with a link to the pseudo-code in the appendix."
>
> Thank you for the suggestion, this has been added in the revised manuscript.
>
> > 10. Reviewer-Nxf3: "The choice of accuracy measure is perhaps not very appropriate for the data: for instance, we expect more drugs to be labeled negatively (for instance, in drug discovery) than positively. Area Under the Curve is said to be slightly more appropriate on imbalanced data."
>
> Indeed, class imbalance is be a persistant challenge in biological applications. The overall framework of hybrid screens and inference set design that we present is not focused specifically at addressing that challenge but different metrics could indeed be employed in acquisition functions and stopping criterion to specialize the framework for such cases.

---

> ### Author Response · Authors · 2024-11-20
> **Author Response to Reviewer #4 (Nxf3) -- (3/3)**
>
> > 11. Reviewer-Nxf3: "Is there a closed-form expression or at least an upper bound on the stopping time in [Equation 4, updated manuscript]"
>
> This is a very interesting question!
>
> There is a trivial upper bound on the stopping time: if we assume a null accuracy on the acquired batch for all remaining steps, i.e. $\mu_b^t=0$ for all $t$, the agent would stop after a number of steps corresponding to our confidence-level for the bound in proportion of the size of the target set, i.e. $\tau = (1-\delta) \cdot |X_{\text{target}}|$. Our budget reduction would thus be of $\delta$ and the system accuracy would be $1-\delta$. To go further than this trivial bound, we would need to make assumptions about the accuracy of the model, which could then allow to upper bound the stopping time more tightly. For exemple we could assume that a particular kind of model can always do at least as good as random chance -- the active acquisition process cannot harm. However, this particular assumption might not hold in practice. For exemple, in general, due to the no-free-lunch theorem, there is no guarantee that we can do better than random (the labels $y$ for a particular task could be completely unrelated to $x$). The model could also happen to be highly uncalibrated. It could then perform worse than random.
>
> Concerning a closed-form solution, if the model was perfectly calibrated, and could provably remain so for the entire duration of the acquisition loop, then it would be possible to derive an expression of the stopping time by precisely predicting what the system-accuracy would be at any step $t$ of the acquisition loop. Unfortunately, modern deep neural networks are known to often be poorly calibrated as shown in [1] and exemplified in our added experiments in Figure 14. This is one of the motivations for relying on the weaker assumption of *weak* calibration (in Lemma 1).
>
> Finally, our experiments in Figure 14 assess the calibration of the models in retrospect, but it would be possible to obtain a calibration estimate for an ongoing active deployment without having access to the labels for the inference set. However, this would require sampling our acquisition batch with a non-zero portion of random samples, which in turn would likely decrease our budget reductions. In future work, it would be interesting to investigate that possibility: evaluate whether accurate calibration estimates can be obtained in live-deployment, and if such estimate could be combined with calibration-improvement methods to allow the agent to predict its stopping time before triggering early in the acquisition loop.
>
> [1]: Guo, C., Pleiss, G., Sun, Y., & Weinberger, K. Q. (2017, July). On calibration of modern neural networks. In International conference on machine learning (pp. 1321-1330). PMLR.
>
> > 12. Reviewer-Nxf3: "At each time $t$, the model has to be retrained and evaluated and then used for selection. How expensive is it in practice?"
>
> For the model architectures that we use in this paper, this is very fast even on single GPU. Ensembles can also be efficiently trained in parallel using einstein summation operators. One of the aspects at play here is that we are considering datasets of thousands or a few million of samples, and not datasets such as text corpora which can include trillions of tokens. Moreover, earlier in the acquisition loop, the observation set $X_{\text{obs}}^t$ is still small, allowing for fast training of the model. Finally, in general, in a real-world active learning deployment, even if we were training a large model on larger corpuses, acquiring a single batch of several thousand samples in wet labs would require days or weeks and can cost tens of thousands or even millions of dollars, making the relative cost and time of training the model for a single acquisition step negligible compared to the costs and time of obtaining that data.
>
> > 13. Reviewer-Nxf3: "Why not a more adaptive approach where all points such that the maximal proba across class labels is no higher than 0.5 (and then the acquisition batch size is no longer a constant)?"
>
> To maximise such investments, large experimental wet labs are often operated at maximum capacity, leading to typically known and fixed experimental batch sizes. For this reason, in this work, we favour acquisition functions that sort the samples according to some score to accomodate this fixed batch size rather than saturation-based methods that would imply a varying acquisition batch-size at different steps.
>
> > 14. Reviewer-Nxf3: "What the “saving” mentioned in Figure 1? The difference between the number of actually measured points and measuring all samples at a given point in time?"
>
> It simply refers to the budget savings (or budget reductions) of the acquisition costs incurred by not having to obtain the labels for the entire target set $|X_{\text{target}}|$ samples: savings = $\frac{|X_{\text{inf}}|}{|X_{\text{target}}|}$, costs = $\frac{|X_{\text{obs}}|}{|X_{\text{target}}|}$.

---

> ### Comment · Reviewer_Nxf3 · 2024-11-21
>
> Thank you for addressing all the reviewers' concerns. The paper is largely improved and I raise my score to 6. I still have a few very minor concerns listed below. Solving them would allow my score to raise to 8 (I would have chosen 7 because the technical tools are not really novel, but 7 is not available).
>
> 1. I have an issue with the notation $P[y=\hat{y} \mid p^\dagger_i]$ in the statement of Lemma 1 on page 4 and in the proof of Lemma 1. If I understood correctly, $P(y=\hat{y} | p^\dagger_i)$ is the probability of assigning class label $\hat{y}$ to sample i given that the probability assigned to its most probable class is $p^\dagger_i$. Then I think the notation  $P[y_i=\hat{y}_i \mid p^\dagger_i]$ (that is, adding the index i) would make this definition clearer. For instance, in the statement of Lemma 1:
>
> $p^\dagger_1 \leq p^\dagger_2 \implies P[y_1=\hat{y}_1 \mid p^\dagger_1] \leq P[y_2=\hat{y}_2 \mid p^\dagger_2]$
>
> There are also both $\hat{p}^\dagger_i$ (with a hat) and $p^\dagger_i$ (without a hat) in the statement and the proof of Lemma 1, but I assume it is the same object?
>
> 2. In the proof of Lemma 1, the following sentence appears: "[...] the samples in $\underline{D_\text{inf}^{t-1}}$ are ordered by confidence where $x_1$ has the predicted class with the lowest confidence and $x_N$ has the predicted class with the highest confidence." But in Equation (11) the samples from the inferred set have indices $N_b+1$, $N_b+2$, ..., $N$.
>
> Also in the proof of Lemma 1, I think it is important to mention that those probability scores are computed over the dataset **at time t** so that they might change across the iterations.
>
> 3. In the caption of Figure 13: how many iterations did you run? Are the error bars standard deviations?
>
> Typos:
>
> Page 10: in the caption of Figure 9: "acquisiton" => "acquisition"
>
> Page 10: end of Section 4/4: "targetted" => "targeted"

---

> > ### Author Response · Authors · 2024-11-21
> > **Author Response**
> >
> > Thank you for the prompt response! We greatly appreciate the reviewer updating their score and continuing the conversation to further improve the clarity of the paper. We have now uploaded a second revision of the manuscript with these additional changes.
> >
> > > 1. If I understood correctly, $P[y=\hat{y}|\hat{p}^\dagger_i]$ is the probability of assigning class label $\hat{y}$ to sample i given that the probability assigned to its most probable class is $\hat{p}^\dagger_i$. Then I think the notation $P[y_i=\hat{y}_i|\hat{p}^\dagger_i]$ (that is, adding the index i) would make this definition clearer.
> >
> > The distinction we attempted to make here was indeed a little too subtle and easy to miss.
> >
> > **To answer your question:** The absence of indices $i$ in $P[y=\hat{y}|\cdot]$ is intentional, as $P[y=\hat{y}]$ is not denoting the probability that a particular prediction $\hat{y}_i$ is equal to its associated label $y_i$. $P[y=\hat{y}]$ is the *expected* probability that the model obtains the correct label across the entire dataset, where we used $y=\hat{y}$ to denote the "event" of correct classification. This is the same as saying that $P[y=\hat{y}]$ represents the accuracy of the model across the dataset. Similarly $P[y=\hat{y}|\tilde{p}]$ or $P[y=\hat{y}|\hat{p}^\dagger_i]$ was then expressing the accuracy of the model conditional to a particular *confidence* of the model, either some generic value $\tilde{p} \in [0,1]$, or for the actual confidence of the model $\hat{p}^\dagger_i$ for a specific sample $x_i$. In short, $\mathbf{1}(y_i=\hat{y}_i)$ (i.e. indicator function and with indices) denotes whether a particular prediction is equal to its label, and $P[y=\hat{y}]$ denotes the expected probability of this same event but across the dataset.
> >
> > **Proposed correction:** We have attempted to clarify these elements in the updated revision in two ways:
> > 1. We give a longer introduction of the notion of calibration and the notation that we use in the text of Appendix A.1 (lines 722-732).
> > 2. We changed the symbol used in the conditionals from $\hat{p}^\dagger_i$ to $\hat{v}_i$ to make it clear that $\hat{v}$ represents a single scalar value, not the model $\hat{p}$ itself. We defined it in the same way as before, i.e. $\hat{v}_i:= \max_k \hat{p}_k(x_i)$, and use it in the probabilities mentioned above as $P[y=\hat{y}|\hat{v}_i]$ -- the probability that the model is correct (across the whole dataset) when the model's confidence is $\hat{v}_i$.
> >
> > > 2. There are also both $\hat{p}^\dagger_i$ (with a hat) and $p^\dagger_i$ (without a hat) in the statement and the proof of Lemma 1, but I assume it is the same object?
> >
> > Yes indeed, this was an error. It has been corrected in the newly updated manuscript.
> >
> > > 3. In the proof of Lemma 1, the following sentence appears: "[...] the samples in $D_{\text{inf}}^{t-1}$ are ordered by confidence where $x_1$ has the predicted class with the lowest confidence and $x_N$ has the predicted class with the highest confidence." But in Equation (11) the samples from the inferred set have indices $N_b+1, N_b+2, \dots, N$.
> >
> > Yes, this is intentional. This distinction stems from the fact that in the former case (Equation 6), we are describing the ordering of sample scores in the inference set at time $t-1$ containing $N$ samples  i.e. $D_{\text{inf}}^{t-1}$. However, in the latter case of Equation 11 we are highlighting the sample-selection process at play, the first `\underbrace` identifies the first $N_b$ terms that belong to the acquisition batch $D_b^t$, and the second `\underbrace` identifies the remaining $N - N_b$ terms which will form the new, *updated* inference set of time $t$, i.e. $D_{\text{inf}}^t$. In summarry, the difference in number of terms from $N$ to $N-N_b$ originates from advancing from one timestep in the acquisition loop: $D_{\text{inf}}^t := D_{\text{inf}}^{t-1} \setminus D_{b}^t$. We have added a clarification to that regard on Line 721.
> >
> > > 4. In the proof of Lemma 1, I think it is important to mention that those probability scores are computed over the dataset at time t so that they might change across the iterations.
> >
> > Indeed. We have added a short mention of this time-dependency of this dependence in the text at Line 716 of the updated manuscript.
> >
> > > 5. In the caption of Figure 13: how many iterations did you run? Are the error bars standard deviations?
> >
> > Thank you for pointing this out. The error bars in the Figure 13 denote minimum and maximum observed system accuracy at stopping time $\tau$ across 3 seeds. This information was added to the caption in the updated manuscript.
> >
> > ----
> >
> > We hope that our answers clarify these remaining ambiguities. Once again thank you for further helping us improving the clarity of the presentation and we remain available for further discussions.

---

> ### Comment · Reviewer_Nxf3 · 2024-11-22
>
> Thank you for the reply.
>
> 1. I am not sure that I get the definition of $P[y=\hat{y}]$. Is $P[y=\hat{y}]$ equal to $P[\forall i \leq N, y_i=\hat{y}_i]$? I would assume that $\hat{p}^\dagger_i$ only impacts the probability of assigning a label to sample $x_i$. But as long the definition of weak calibration is the one actually written in the paper, the lemma remains true.
>
> I am not sure that changing $\hat{p}^\dagger_i$ for $\hat{v}_i$ is a good idea. A reader might wonder where $v$ comes from. I would rather suggest either use $P[\forall i \leq N, y_i=\hat{y}_i]$ or put in bold the $y$ and $\hat{y}$ so that it is clear that they are vectors.
>
> 2. 3. 4. The changes brought to the paper answer my concerns.
>
> 5. Since Lemma 1 should fail in practice 5% of the time, only 3 iterations in Figure 13 do not satisfyingly illustrate the lemma. The best would be N=100.

---

> ### Author Response · Authors · 2024-11-25
> **Author Response (1/2)**
>
> Thank you for your feedback and suggestions. We have uploaded a third revision of our manuscript and address here each point separately:
>
> ---
>
> ### About the notation
>
> > 1. I am not sure that changing $\hat{p}^\dagger_i$ for $\hat{v}_i$ is a good idea. [...] I would rather suggest either use $P[\forall i \leq N, y_i=\hat{y}_i]$ or put in bold the $y$ and $\hat{y}$ so that it is clear that they are vectors.
>
> To denote the confidence value of the model (the max probability over classes for some sample $x_i$), we still believe that changing from $\hat{p}^\dagger_i$ to $\hat{v}_i$ can help avoid some elements of confusion it might help the reader distinguish the between $\hat{p}$ (the model itself) and this scalar confidence *produced* by the model. This is important because the probabilities $P[y=\hat{y}]$ are always taken w.r.t the model $\hat{p}$, but could also be conditioned with any given scalar confidence value $\tilde{v} \in [0,1]$ which are not necessarily produced by the model.
>
> For the probability $P[y=\hat{y}]$, we use the notation from [1], which is a standard source for calibration analysis in deep neural networks. Although this notation carries useful semantic information (it can be read as "the probability that the model is correct on any given prediction") we agree that this notation is somewhat loose, both because (1) it implies only very subtly (by the absence of index $i$) the notion that several samples are involved in this quantity (it is an expectation), and (2) because it does not specify explicitly which dataset is involved in this expectation. In our work, there is indeed one important difference with [1] which would indeed justify slightly adjusting the notation: in [1], it was sufficient to keep the dataset implicit because every experiment only contained a single dataset. However, in our active learning setting which makes use of different partitions for the observation set, the inference set, the acquired batch, and so on, it would be useful to be more explicit. We have thus attempted to further improve our notation for $P[y=\hat{y}]$ in the following way:
>
> - We changed the content of the bracket for capital letters, i.e. $P[y=\hat{y}]$ has become $P[Y = \hat{Y}]$. This is exactly how this quantity is denoted in [1] so it might be even more familiar to the reader and it better distinguishes the probability of the model being correct by referring to the random variable $Y$ from a specific instantiation of this variable $y_i$.
> - We explicitly specify the dataset $D$ over which the expectation is computed, changing from $P[Y=\hat{Y}]$ to $P_{D}[Y=\hat{Y}]$. Therefore, we can see exactly which dataset is used in the expectation, e.g. $P_{D_b^t}[Y=\hat{Y}]$ for the acquisition batch, $P_{D_{\text{inf}}^t}[Y=\hat{Y}]$ for the inference set.
>
> [1]: Guo, C., Pleiss, G., Sun, Y., & Weinberger, K. Q. (2017, July). On calibration of modern neural networks. In International conference on machine learning (pp. 1321-1330). PMLR.*
>
> ### About the definition of $P_D[Y=\hat{Y}]$
>
> > 2. I am not sure that I get the definition of $P[y=\hat{y}]$. Is $P[y=\hat{y}]$ equal to $P[\forall i \leq N, y_i=\hat{y}_i]$?
>
> As mentioned above, $P_D[Y=\hat{Y}]$ is an expectation over the indicator function indicating if the model's prediction is correct for all samples within $D$. Formally, $P_D[Y=\hat{Y}]:=E_{(x_i,y_i)\sim D}[\mathbf{1}(y_i = \hat{y}_i)]$. It can be equivalently thought of as the accuracy of the model over the dataset $D$, or as "the probability that the model is correct" for any sample $x_i$ uniformly drawn from $D$. In the revised manuscript, in addition of pointing the reader to [1], we now fully state its definition and its confidence-conditional equivalent in Equations 7 and 8. We also use this more compact description for for the accuracy in Lemma 1.

---

> > ### Author Response · Authors · 2024-11-25
> > **Author Response (2/2)**
> >
> > With these explanation and improved notation, we can address this question:
> >
> > > 3. I would assume that $\hat{p}^\dagger_i$ (now $\hat{v}_i$) only impacts the probability of assigning a label to sample $x_i$.
> >
> > No this is incorrect. In its *un*conditional, i.e. $P_{D}[Y=\hat{Y}]$, the expectation is taken over the entire dataset. In its *conditional* form, i.e. $P_{D}[Y=\hat{Y}|\tilde{v}]$, the expectation is taken over all the samples in $D$ for which the confidence of the model is equal to $\tilde{v}$. If $\tilde{v}=0.75$, then $P_{D}[Y=\hat{Y}|\tilde{v}]$ represents the accuracy of the model over all the samples for which the model has a confidence of $0.75$. When we denote the conditional with the confidence $\hat{v}_i $ for a specific exemple $x_i $, i.e. $ P_D[Y=\hat{Y}|\hat{v}_i] $, it simply means that we are interested in the accuracy of the model on $x_i$ *and* all the other samples for which the model's confidence is also equal to $\hat{v}_i$. As noted in the paper, in practice we are dealing with finite datasets and no sample will yield the *exact* same confidence value, so such confidence levels would be binned together such that $P_D[Y=\hat{Y}|\tilde{v}]$ represents the accuracy over all samples in $D$ for which the model's confidence falls in $[\tilde{v} - \epsilon, \tilde{v} + \epsilon]$.
> >
> > ### About the empirical validation of our stopping criterion
> >
> > > 4. Since Lemma 1 should fail in practice 5% of the time, only 3 iterations in Figure 13 do not satisfyingly illustrate the lemma. The best would be N=100.
> >
> > We indeed agree that because of its probabilistic nature, 3 seeds were not sufficient to robustly verify that the bound we present in Equation 5 holds in practice. We initially produced this analysis from the data resulting from our other experiments in the paper, which were run on 3 seeds because the standard error (shaded area) for all curves was often very low and did not require more trials to support our claims.
> >
> > To increase the robustness of this analysis we have run both the least-confidence based agent and the random sampling baseline for all five classification experiments with 100 different random seeds each for the MNIST variations and QM9, and 50 seeds for the more computationally expensive phenomics experiments. Please find these updated results in Appendix F.2, Figure 14 of the updated manuscript. The results show that in all experiments with the least-confidence agent, the system accuracy was above the targeted threshold $\gamma$ when our proposed stopping criterion induced the agent to stop acquiring new labels. For the random agents, the occurrences of bound failure (i.e. where the measured system accuracy was below the threshold at stopping time was respectively 1%, 0%, 0%, 6% and 0%). See Figure 14 in the updated manuscript for these results.
> >
> > There are two important considerations for interpreting these results:
> >
> > 1. Our estimator $\alpha_t$ from Equation 5 for the inference set accuracy does not imply that when using $\delta=0.05$, we should expect the true inference set to be smaller than our estimate exactly 5% of the time (i.e. that we have an equality with $\delta$), instead, $\delta$ is an upper-bound on the probability of observing that even, i.e. $P(\mu_{{\text{inf}}}^t \leq \alpha_t) \leq \delta$. Chernoff bounds are sometimes more conservative than necessary (the bound is often not tight) and therefore the true probability of failure for some experiments might be quite lower than $\delta$. Therefore, even for the random agent, it is not contradictory to observe that for some experiments 100 out of 100 trials lead to bound success.
> > 2. As shown in Lemma 1, for the least-confidence based agent, this bound in Equation 5 is even more conservative since our estimate for the inference set accuracy -- the accuracy on the batch -- is *itself* a lower-bound estimate. Therefore, the margin between the true system accuracy at stopping time and its targeted threshold is likely to be even greater, and this is what is indeed observed by comparing the LC agent and the Random agent in our updated analysis in Figure 14. This observation brings us back to the main takeaway that we wish to communicate in this work: while the LC agent's estimate of the inference set accuracy is more conservative than the Random agent, it still is able to stop acquisition far sooner than the Random agent because its acquisition function is strategically designing the composition of the inference set to allow the system to reach high performance on this particular dataset, thus compensating many times over its looser bound in the stopping criterion. We briefly touch this point in the final discussion in Section 5.
> >
> > ---
> >
> > We hope that our answers clarify these remaining ambiguities. Once again we thank the reviewer for engaging so productively during this discussion period and firmly believe that these iterations have lead to significant improvements in the paper.

---

> > > ### Comment · Reviewer_Nxf3 · 2024-11-25
> > >
> > > Thanks for taking the time to address my concerns. Everything has been solved to me. I raise my score to 8.
> > >
> > > Typo:
> > > - Missing reference on Line 508: "We also empirically validate (see Figure ??, Appendix F.2)".

---

> > > > ### Author Response · Authors · 2024-11-25
> > > > **Author response**
> > > >
> > > > Thank you for the typo, this has been corrected in Overleaf and will be part of the next manuscript revision (holding for now to avoid cluttering).

---

### Official Review · Reviewer_VGSj · 2024-10-25

**Soundness:** 2
**Presentation:** 2
**Contribution:** 2
**Rating:** 5
**Confidence:** 3

**Summary:**

This paper tackles the challenge of reducing experimental costs in high-throughput biological screening, with a focus on real-world applications such as drug discovery and cell microscopy. To achieve this, the authors formulate it as a sequential subset selection problem, which seeks to identify the smallest possible subset of compounds for experimental testing, whose outcomes can ensure a desired level of accuracy for the entire system. In other words, the authors propose a mechanism that aims to actively identify and experimentally test the most challenging compounds, leaving the relatively easier-to-predict compounds for the inference set. To strategically label only the most challenging compound samples, the authors propose to employ an uncertainty-based active learning approach that selects samples for which the model shows the lowest predicted probabilities or where ensemble models exhibit the highest disagreement.

**Strengths:**

The strength of this paper is derived from addressing an interesting real-world challenge in biological screening, where experimental costs are one of the most significant bottleneck. The proposed inference set design method shows somewhat promising potential for these applications by enabling a strategically targeted selection of samples based on model uncertainty. Another strength is the use of diverse real-world datasets, ranging from molecular property prediction (QM9 and Molecules3D) to cell microscopy experiments (RXRX3 dataset with 17,063 CRISPR knockouts and 1,674 FDA-approved compounds).

**Weaknesses:**

The most significant weaknesses of this paper are both technical and presentational. First, this paper does not meet basic ICLR submission guidelines, as its length exceeds the 10-page limit, and this paper contains formatting issues, including a large blank space on page 5 and broken text formatting on lines 392-393 of page 8, which diminishes its professional presentation. Secondly, the experimental comparison is severely limited, with only a random agent used as a baseline method. Given the paper's relevance to active learning and sample selection, the absence of comparisons with reinforcement learning (RL) approaches or other active learning methods is a significant shortfall. Specifically, an RL agent with proper reward modeling could potentially offer competitive performance in sample subset selection. Lastly, the theoretical explanation of Lemma 1 lacks clarity.

**Questions:**

1. What is $N_{\text{target}}$ on page 2? The notation appears without explanation, leaving its meaning unclear.
2. In Lemma 1, the inequality $P[\hat{Y} = Y \mid \hat{p} = p_1] > P[\hat{Y} = Y \mid \hat{p} = p_1]$ is used, but since both sides are identical, the inequality does not make logical sense.
3. In the proof of Lemma 1, the notation $i \in\in\mathcal{X}^{t}_{\text{inf}}$ is unclear. Additionally, the step claiming “the inequality follows from weak calibration and least confidence’s selection” does not seem justified, as it appears to have logical gaps and doesn't rigorously establish the claimed inequality. The overall proof structure needs a more detailed explanation to establish the claimed result.
4. On page 4, lines 212 to 213, this assertion seems to be logically circular, and the abbreviation 'IID' is used without definition. Although 'IID' is a common term, it should be defined in full upon its first appearance in the paper.
5. On page 6, the authors distinguish their approach from Bayesian methods but only compare it with a random agent. However, established Bayesian methods like Bayesian Active Learning by Disagreement (BALD) [1] can effectively select samples that are expected to reduce uncertainty across the dataset. The authors should explain why Bayesian active learning methods cannot be applied in this context and clarify their decision not to compare their approach with BALD or other active learning or reinforcement learning methods.
6. If the method's effectiveness is constrained by a 'low data regime and complex task' (as indicated by the RxRx3 dataset results), how can it be practically useful for real-world biological applications that commonly encounter these challenges?




[1] Houlsby, Neil, et al. "Bayesian active learning for classification and preference learning." arXiv preprint arXiv:1112.5745 (2011).

---

> ### Author Response · Authors · 2024-11-20
> **Author Response to Reviewer #3 (VGSj) -- (1/2)**
>
> We wish to thank Reviewer-VGSj for their thorough review which we believe has allowed to significantly improve the paper. We now aim at addressing specific claims that were raised:
>
> > 1. Reviewer-VGSj: "What is $N_{target}$ on page 2?"
>
> It is simply the number of samples that constitute the target set of interest $N_{target}:=|X_{target}|$. In the framework that we promote, and which we argue is highly relevant to real-world applications, experimenters seek to obtain readouts $y$ for a target set of samples $x$. Inference set design powered by a least-confidence acquisition function allows to obtain provably accurate predictions for the inference set, which constitute the portion of the target set that hasn't been acquired by the agent when the stopping criterion is triggerred, i.e. $X_{\text{inf}}:= X_{\text{target}} \setminus X_{\text{obs}}$. These explanations have been clarified (see Section 2, updated manuscript).
>
> > 2. Reviewer-VGSj: "the theoretical explanation of Lemma 1 lacks clarity [...] The overall proof structure needs a more detailed explanation to establish the claimed result."
>
> We agree that the theoretical section was lacking clarity and contained important notational typos, thank you for pointing this out. We have corrected it by improving the notation and explanations surrounding our theoretical contributions (see Section 2.2, updated manuscript) and we have added in appendix a detailed derivation for both the proof of Lemma 1 and the derivation of the bound to further support the main text (see Appendices A.1 & A.2, updated manuscript).
>
> Note that the essense of Lemma 1 and the bound consituting our stopping criterion remain exactly the same, but we believe their presentation has been greatly improved.

---

> ### Author Response · Authors · 2024-11-20
> **Author Response to Reviewer #3 (VGSj) -- (2/2)**
>
> > 3. Reviewer-VGSj: "the experimental comparison is severely limited, with only a random agent used as a baseline method. [...] Bayesian Active Learning by Disagreement (BALD) can effectively select samples that are expected to reduce uncertainty across the dataset"
>
> Thank you for the suggestion. We have added BALD in our set of comparisons. The results indeed show that BALD in some cases yields similar benefits as the least-confidence acquisition function (see Figures 5, updated manuscript). This is perfectly in accordance with our thesis -- we believe that hybrid screens and inference set design represent a more promising framework to observe real-world impact from active learning methods than strictly focusing on improvements in generalisation, and we show this across a wide variety of tasks. By performing similarly than least-confidence acquisition on QM9, BALD indeed supports the observation that other appropriately designed acquisition functions are also amenable to the same kind of benefits.
>
> However, it is also important to highlight that our theoretical contributions in Lemma 1 and in the derivation of a bound on the inference set accuracy (see Section 2.2, updated manuscript) assume the use of a least-confidence acquisition function, and therefore we cannot garantee that other acquisition functions will perform similarly to least-confidence in all settings. In our results on phenomics data, we find that BALD performs better than random sampling but not as well as least-confidence on the inference set.
>
> Overall, we believe BALD was an important addition to our baselines and opens up the question of which other acquisition functions (apart from least-confidence) provably leads to high performance levels on the inference set.
>
> > 4. Reviewer-VGSj: "If the method's effectiveness is constrained by a 'low data regime and complex task' (as indicated by the RxRx3 dataset results), how can it be practically useful for real-world biological applications that commonly encounter these challenges?"
>
> This is indeed an important question. One situation where inference set design can still be useful here is if the data regime is low, but the mapping $x\rightarrow y$ is challenging for only *some* samples. Then, assuming weak calibration, an active learning model can help us identify *which* samples are indeed more challenging and eliminate them from the inference set by acquiring their labels. In that sense, inference set design can still be useful on small datasets. However, for a task where we are in a low-data regime and the mapping $x\rightarrow y$ is challenging for *all* samples, then there are no silver bullet and aside of using pretrained models that benefit from useful priors derived on other datasources, the only solution is to acquire more data. Luckily however, there is an increasing number of high-throughput biological screening assays capable of generating thousands of datapoints per week and allowing to apply modern active learning methods at scale [1-4].
>
> [1]: Shalem, O., Sanjana, N. E., Hartenian, E., Shi, X., Scott, D. A., Mikkelsen, T. S., Heckl, D., Ebert, B. L., Root, D. E., Doench, J. G., & Zhang, F. (2013). Genome-Scale CRISPR-Cas9 Knockout Screening in Human Cells. Science, 343(6166), 84–87. https://doi.org/10.1126/science.1247005
>
> [2]: Mahjour, B., Shen, Y., & Cernak, T. (2021). Ultrahigh-Throughput Experimentation for Information-Rich Chemical Synthesis. Accounts of Chemical Research, 54(10), 2337–2346. https://doi.org/10.1021/acs.accounts.1c00119
>
> [3]: Celik, S., Hütter, J., Carlos, S. M., Lazar, N. H., Mohan, R., Tillinghast, C., Biancalani, T., Fay, M. M., Earnshaw, B. A., & Haque, I. S. (2022). Biological Cartography: Building and Benchmarking Representations of Life. bioRxiv (Cold Spring Harbor Laboratory). https://doi.org/10.1101/2022.12.09.519400
>
> [4]: Blay, V., Tolani, B., Ho, S. P., & Arkin, M. R. (2020). High-Throughput Screening: today’s biochemical and cell-based approaches. Drug Discovery Today, 25(10), 1807–1821. https://doi.org/10.1016/j.drudis.2020.07.024

---

> > ### Comment · Reviewer_VGSj · 2024-11-25
> > **Reviewer Response**
> >
> > I have thoroughly reviewed the rebuttals and related discussions, and I have decided to raise my score from 1 to 5. This is an interesting topic; however, while the authors have added BALD for additional baseline methods, they did not mention reinforcement learning methods, which I previously raised as a concern. Furthermore, I feel that the number of baseline methods is quite low, and the technical novelty remains somewhat limited.

---

> > > ### Author Response · Authors · 2024-11-25
> > > **Author response (1/2)**
> > >
> > > Thank you for reviewing the updated mansucript and for updating your score. In this response we would like to clarify the remaining concerns.
> > >
> > > ---
> > >
> > > ### About the absence of an RL-based approach
> > >
> > > > 1. The authors did not mention reinforcement learning methods, which I previously raised as a concern.
> > >
> > > We appreciate the suggestion to explore an RL approach. However, we deliberately chose to focus on AL methods because this paradigm is more directly suited to the task of reducing experimental budget. While RL could potentially be applied in a similar context, there are several reasons why this is not a straightforward setting in which to deploy an RL solution:
> > >
> > > 1. **Large observation and action spaces**: Employing RL here would require carefully defining the observation and action spaces with which the agent would be interacting. One option would be to let the agent observe the entire inference set $X_{\text{inf}}^t$ at each time-step $t$. This could be realised using a policy parameterised by deep-set models or a transformer, and define the action space as the set of indices for these samples. However this parameterised policy would significantly increase the computational complexity of the method and drastically complexify the learning interactions between the inner-loop and the outer-loop of the active learning sequence.
> > > 2. **Reward misspecification with dense reward functions**: Another major design decision is that of the reward function. Reinforcement learning is notoriously prone to reward mispecification, often causing the agent to behave in a way that was unintended and misaligned with the user's intentions [1-5]. Here the goal is to obtain a model which can stop acquiring labels early while performing well on the inference set. The only estimate that we can have of the inference set comes from the acquired batch. A *dense* reward function in this problem could reward the policy for performing well on the acquired batch, hoping that this would be reflected in the inference set. However a simple exploitative policy to maximise such a signal is to only pick the *easiest* samples for acquistion, which would have the opposite effect on the inference set: leaving it with the most difficult examples. Methods like bandit algorithms, which we discuss in our Related Work, would require a such dense reward function, but it is not readily available in our setting.
> > > 3. **Sample inefficiency with sparse reward functions**: A better alternative than to design a dense reward proxy could be to only reward it at the end of the acquisition phase. However RL algorithms are known to be highly sample-inefficient and a sparse reward scheme would exacerbate the issue [6]. The challenge here is that this agent must be trained on the same data that it would be deployed on. If we were interested in acquiring a *large* number of such target sets $X_{\text{target}}$, *some* proportion of them could be used as training episodes for an RL agent which could be deployed on the rest. However, in the setting that we present here, where we wish to realise gains even for the acquisition of a *single* target set, training the policy with a sparse reward is not feasible.
> > > 4. **Hyperparameter tuning**: RL methods such as TD3 and PPO often involve a large number of hyperparameters which need to be adjusted from one task to another [7]. In contrast, the AL methods we employ can be applied "out-of-the-box" with minimal tuning as we demonstrate across diverse set of experiments.
> > > 5. **Lack of theoretical garantees**: Finally, in the paper, we provide strong theoretical justifications for using least-confidence based acquisition functions (see Section 2.2 and Appendix A), and validate that our theoretical assumptions and results hold in practice (see Appendix F). The use of a learned RL policy for sampling acquisition batches would invalidate these theoretical garantees.
> > >
> > > In conclusion, while Reinforcement Learning is a powerful paradigm for learning in settings involving a known reward function (e.g. Go), in the problem tackled in this paper, we are fundamentally in a supervised framework and active learning algorithms represent a much more natural set of methods for real-world deployments.
> > >
> > > [1]: Amodei et. al. (2016). Concrete problems in AI safety.
> > >
> > > [2]: Gupta et. al. (2022). Unpacking reward shaping: Understanding the benefits of reward engineering on sample complexity.
> > >
> > > [3]: Knox et. al. (2023). Reward (mis) design for autonomous driving.
> > >
> > > [4]: Pan, et. al. (2022). The effects of reward misspecification.
> > >
> > > [5]: Dewey, D. (2014). Reinforcement learning and the reward engineering principle.
> > >
> > > [6]: Ng et. al. (1999). Policy invariance under reward transformations: Theory and application to reward shaping.
> > >
> > > [7]: Henderson et. al. (2018). Deep reinforcement learning that matters.

---

> ### Author Response · Authors · 2024-11-25
> **Author Response (2/2)**
>
> ### About the number of baselines
>
> > 2. While the authors have added BALD for additional baseline methods [...] I feel that the number of baseline methods is quite low.
>
> We agree that having a sufficient and appropriate set of baselines is of great importance for proper experimental validation. Once again we thank the reviewer for suggesting to add more such baselines to the paper, in reaction to which we added *BALD* (specifically requested by the reviewer) and a *diversity-based agent*, a very standard baseline in AL and belonging to a complementary family of methods [1]. We also include a random sampler, often found to be a surprisingly strong baseline [2,3], and heuristic-based orderings, which are probably the most commonly employed acquisition strategy in the industry. Including all of these, in the current state of the paper, we thus present results on 9 different acquisition functions across 7 different datasets spanning both classification and regression tasks. While it is always possible to add more, ever extending the set of comparisons faces diminishing returns as existing AL benchmark studies often find comparable performance across many active learning strategies [4,5]. We acknowledge that more sophisticated acquisition functions could potentially yield marginal improvements on the stopping time, but these improvements are orthogonal to the inference set design perspective and the core findings of our study.
>
> When considering adding more experiments, the central question should be whether these experiments would allow to better support specific claims made in the paper. We would like to take the opportunity to summarize the different claims we make and how we support these claims with appropriate experiments:
> 1. We claim, in Sections 1,2,5, that active learning focused only on generalisation improvements on a held-out test set obscures the practical benefits that AL methods can yield in terms of experimental cost reduction, and that focusing on a finite target set can yield significant cost reduction improvements and represents a more robust approach to real-world AL deployment.
>     - We support this claim throughout all of our experiments in Section 4 by showing (1) the absence of a consistent gap between active agents and a random samplers on the test accuracy, alongside (2) important performance gaps between active agents and random samplers on the inference set accuracy (see Figures 2, 3, 5, 8, & 10).
> 2. We claim, in Sections 1,2,5, that the mechanism responsible for this improvement, which we call *inference set design*, consists in acquiring the most difficult examples first such that the inference set is left with the relatively easier examples to be evaluated on, and through Lemma 1 claim that assuming weak calibration of the model this can be achieved by using a least-confidence based acquisition function.
>     - We provide a mathematical proof for the correctness of Lemma 1 in Appendix A.
>     - We empirically validate that the assumption of Lemma 1 holds in practice in Appendix F.1.
>     - We empirically validate the inference set design mechanism by tracking and analysing which samples are acquired first by the agents in Figures 4, 6 and 11.
> 3. We claim in Section 2.2, Equation 5, that a Chernoff bound on the acquisition batch accuracy can be used to obtain an effective stopping criterion for this procedure.
>     - We validate experimentally by measuring the stopping time $t=\tau$ and the associated cost reductions that our stopping criterion combined with a least-confidence agent allows experimenters to realise important cost reductions (see stopping times in Figures 2, 5, 8 & 10).
>     - We validate that the bound failure limit $\delta$ specified in this theoretical bound is respected in practice (see Appendix F.2).
>
> For the reasons highlighted above, we believe that our set of baselines and experiments appropriately supports the claims made in the paper, and that this perspective can bring significant value to experimenters working at the intersection of active learning and real-world applications.
>
> [1]: Wu, D. (2018). Pool-based sequential active learning for regression.
>
> [2]: Yang, et. al. (2018). A benchmark and comparison of active learning for logistic regression.
>
> [3]: Guo, et. al. (2022). DeepCore: A Comprehensive Library for Coreset Selection in Deep Learning.
>
> [4]: Margatina, et. al. (2021). Active Learning by Acquiring Contrastive Examples.
>
> [5] Trittenbach, et. al. (2020). An overview and a benchmark of active learning for outlier detection with one-class classifiers.
>
> ---
> We hope these clarifications address your concerns. Thank you once again for the detailed feedback provided on this work, we believe that it has lead to significant improvements of the proposed manuscript! We remain available until the end of the discussion period for any additional questions.

---

### Official Review · Reviewer_i8MT · 2024-11-02

**Soundness:** 3
**Presentation:** 3
**Contribution:** 3
**Rating:** 6
**Confidence:** 4

**Summary:**

The paper presents an active learning method for hybrid screening, for the setting where there is a finite library and the goal is performance on this library (and not generalization). The idea is to use an active learning approach to include the difficult examples in the training set and stop when the performance is beyond some threshold. The paper discusses two reasonable acquisition functions for this purpose. In each round, a batch of difficult observations are labelled, and the performance on this batch is used to lower bound the performance on the remaining data in the inference set. The method is demonstrated on several relevant benchmark data sets.

**Strengths:**

The paper is generally well written and easy to follow.

The main idea is clear, useful and effective.

Good experiments to demonstrate the benefits of the approach.

**Weaknesses:**

The technical novelty is limited, as it is mostly an application of existing methods.

The paper would benefit from including comparisons with other heuristics that capture the "difficulty" of observations. For molecules, one could think of many ways to heuristically score their complexity, and use this to train on most complex examples first. I think this would be of practical interest to know how much the proposed method can improve over such baselines.

The paper appears to have formatting issues, with a mostly blank page and a full page of text exceeding the page limit.

**Questions:**

What are key differences to existing uncertainty based active learning methods? It appears to me, that the main difference is that the evaluation criterion is performance on the library and not generalization.

In what practical sense does the stopping criterion provide theoretical guarantees? The paper argues that there is a lower bound on target accuracy, assuming the model is weakly calibrated (i.e., the order of uncertainty is correct). However, how reliably does this assumption hold in practice?

For the committee approach, why do you focus on the voting entropy, rather than e.g. entropy of the mean predictive distribution?

I did not find a description in the main text of the ML models used in the experiments.

If I understand correctly, on Molecules3D you include a comparison with data ordered by molecule size, smallest first. Does this not go against the idea of training on difficult examples first? Would it not be more interesting to compare with data ordered by decreasing molecule size?

Notation:

In eq. 3 Nb is not defined, and the notation is not clear. The left hand side has no index i inside the brackets, and the right hand side evaluates to a single value, not a set. Same goes for eq. 4.

Minor comments:

This sentence is unclear to me: "Because of this, we can treat performance on X_inf^t as though they were IID samples from X_inf^t, and construct a (conservative) lower bound on the accuracy on X_inf.

I am not sure why the method is called "inference set design" - it does not explicitly design the inference set, but is rather a fairly standard active learning scheme with a particular focus on optimizing system performance by including difficult examples in the observation set.

---

> ### Author Response · Authors · 2024-11-20
> **Author Response to Reviewer #2 (i8MT) -- (1/2)**
>
> We thank Reviewer i8MT for their constructive comments on the submitted work as well as for their ideas for improvement. We have addressed all the minor concerns and clarifications that were identified. We now aim at addressing each comment individually:
>
> > 1. Reviewer-i8MT:"The paper would benefit from including comparisons with other heuristics that capture the "difficulty" of observations. For molecules, one could think of many ways to heuristically score their complexity, and use this to train on most complex examples first."
>
> This is an interesting baseline idea. Indeed, real-world drug discovery pipelines often make use of simple heuristics to order or prioritise which samples should be acquired first. We have run additional experiments using the SA-score as a heuristic-based acquisition rule for the molecules (see Figure 5, Section 4.2, updated manuscript). While the SA-score is typically employed as a (loose) indicator of synthesizability, it is essentially a complexity indicator [1]. In our experiments on QM9, we find that using it as acquisition function yields a system accuracy inferior to that of a random sampler, and find similar results for other heuristics such as ordering the molecules from small-to-large, or from large-to-small (see Figure 10, Appendix E.1, updated manuscript). These results illustrate that seemingly intuitive heuristics can harm the performance of the model and that a random sampler, often found to be a surprisingly strong baseline for active learning tasks [2], should generally be preferred to such manual approaches.
>
> [1]: Coley, C. W., Rogers, L., Green, W. H., & Jensen, K. F. (2018). SCScore: synthetic complexity learned from a reaction corpus. Journal of chemical information and modeling, 58(2), 252-261.
>
> [2]: Yang, Y., & Loog, M. (2018). A benchmark and comparison of active learning for logistic regression. Pattern Recognition, 83, 401-415.
>
> > 2. Reviewer-i8MT:"It appears to me, that the main difference is that the evaluation criterion is performance on the library and not generalization."
>
> This is correct, the main acquisition function that we use is standard (least confidence acquisition function). However, the main contribution of this work is to raise awareness about the importance of the evaluation setting of active learning methods and to bring attention to the more useful and applicable scenario of inference set design as opposed to purely focusing on generalisation improvements, which are known to be challenging and subject to hallucinations. Our proposed evaluation setting exacerbates the importance of having an explicit stopping criterion to minimize the number of experiments that are perform. We thus also propose a theoretically grounded stopping criterion to achieve this. Finally, we perform exhaustive empirical evaluations to illustrate the phenomenon of modest improvements on generalisation side-by-side with important gains from inference set design on the inference set.
>
> By additionally experimenting on BALD as baseline (as requested by Reviewer VGSj), we now also show that inference set design can be achieved using other acquisition functions. However, we do not claim novelty on the acquisition functions themselves. To clarify this, we have removed the subsection dedicated to acquisition functions in our methods section, and only introduce them when needed later in the paper (see Sections 2.2 and 4.2, updated manuscript).
>
> > 3. Reviewer-i8MT:"In what practical sense does the stopping criterion provide theoretical guarantees? The paper argues that there is a lower bound on target accuracy, assuming the model is weakly calibrated (i.e., the order of uncertainty is correct). However, how reliably does this assumption hold in practice?"
>
> This is an excellent question. Please take a look at the revised theoretical material which now explains the derivation of the bound in more detail (Section 2.2 & Appendix A, updated manuscript). We have also added an empirical analysis which confirms (1) that the assumption of weak calibration of the model holds in practice for all of our experiments, and (2) that when the algorithm triggers the stop of the acquisition process, the system accuracy is indeed above the desired threshold $\gamma$ (see Section 4.4 and Appendix F, updated manuscript).

---

> ### Author Response · Authors · 2024-11-20
> **Author Response to Reviewer #2 (i8MT) -- (2/2)**
>
> > 4. Reviewer-i8MT:"For the committee approach, why do you focus on the voting entropy, rather than e.g. entropy of the mean predictive distribution?"
>
> This was a clarity error on our part. We were introducing the voting-based query-by-committee (QBC) as an exemple of commonly employed acquisition functions for classification tasks. However, in our experiments, we used QBC only for the regression task on Molecules3D, which does not use neither votes nor class-probabilities but instead computes the variance over the (scalar) predictions made by the ensemble members. Our framework and theoretical contributions is primarily centered around the least-confidence acquisition function. We clarified this by removing the QBC equation from our Methods section (see Section 2, updated manuscript). Note however that we do not claim that the inference set design is exclusively achievable using least-confidence acquisition functions. We have added BALD [3], which is similar to QBC in that it uses an ensemble, as an additional baseline and show that in some cases it obtains similar gains as the least-confidence acquisition function on the inference set (see Figures 5 & 8, updated manuscript).
>
> [3]: Houlsby, Neil, et al. "Bayesian active learning for classification and preference learning." arXiv preprint arXiv:1112.5745 (2011)
>
> > 5. Reviewer-i8MT:"I did not find a description in the main text of the ML models used in the experiments."
>
> For all of our experiments, we used vectorized representations as input and our models are parameterized using a deep feed-forward neural network with residual connections. This was mentioned in Appendix but we now also clarify it in the opening statement of the experimental section (see Section 4, updated manuscript).
>
> > 6. Reviewer-i8MT:"If I understand correctly, on Molecules3D you include a comparison with data ordered by molecule size, smallest first. Does this not go against the idea of training on difficult examples first? Would it not be more interesting to compare with data ordered by decreasing molecule size?"
>
> Yes, in our initial set of experiments the heuristic ordering of molecules in the Molecules3D dataset was from small to large. When adding this heuristic we considered the difference in computational cost of labeling different molecules with Density Functional Theory (DFT). The cost of computing molecular properties using DFT scales as $\mathcal{O}(n^3)$ with the number of electrons in the system (approximately as $\mathcal{O}(N^3)$ with the number of atoms). Therefore, in real-world applications we want to minimize the number of large molecules that have to be computed and labeled with DFT. However, it is often true that predicting properties for larger molecules is harder compared to small molecules. In the updated version of the paper we added a baseline where molecules are acquired from large to small (see Figure 10). We observe that it led to a worse performance especially for total energy predictions because some of the chemical elements that are present in small molecules are missing in large molecules, and therefore in training data. For more details on this analysis please see Section E.1 in the Appendix.
>
> > 7. Reviewer-i8MT:"I am not sure why the method is called "inference set design" - it does not explicitly design the inference set, but is rather a fairly standard active learning scheme with a particular focus on optimizing system performance by including difficult examples in the observation set."
>
> The algorithm implicitly "designs" the inference set by *selecting* the set of examples that composes it. It thus designs the *composition* of the set, and not the samples themselves. When we stop acquiring new labels, what is left in the inference set is crucial since these are the samples for which we will rely on model predictions rather than ground truth observations -- we rely on the model being highly accurate on these samples. In the paper, we show both through Lemma 1 and empirically that a provably high accuracy on that set can be achieved assuming weak calibration of the model. This is very different from standard active learning where generalization error is unaffected by the choices that the algorithm makes except through its effect on model quality. If the target set of potential examples was infinitely large, inference set design would reduce to standard active learning (because with an infinitely large inference set, the choices that the learner makes do not affect the composition of the inference set), but in the finite library setting, performance can be improved significantly by leaving only the relatively easier examples in the inference set.

---

> > ### Comment · Reviewer_i8MT · 2024-11-22
> >
> > Thank you for your detailed response and for the additional experiments you have added to the manuscript. I maintain my rating, and recommend the paper be accepted. I think the paper is solid and interesting, although the technical novelty is somewhat limited.

---

### Official Review · Reviewer_Et7D · 2024-11-04

**Soundness:** 4
**Presentation:** 4
**Contribution:** 3
**Rating:** 6
**Confidence:** 4

**Summary:**

The paper describes the application of active learning for Inference Set Design, a problem where the goal is to label a predefined finite set of instances. To this end, a hybrid screen is proposed where experiment carried out on a subset of the instances (the observation set) and the label on the rest of instances (inference set) is approximated by the prediction of a model trained on the observation set. The paper shows that in this setup the inference set design aspects dominate the better model generalization aspects of active learning.

**Strengths:**

The described problem is very close to industrial application, and as such quite relevant.  The paper is clearly written and quite easy to follow. The experiments are quite detailed.

**Weaknesses:**

The machine learning novelty is limited, it describes classical active learning evaluated with a non traditional performance metric. (This metric is however quite well motivated). The argument by the authors, that the benefit of active learning should be interpreted more broadly if the goal is to label the dataset is valid. Using similar argument, however, most of the time the final goal is to maximize (target) or minimize (off-target, tox.) the outcome of the experiment (at least find multiple diverse extrema), the problem become a Bayesian optimization problem, with already existing literature and efficient methods. In the light of this, can you point out a few practical scenarios where ISD is still relevant?

The paper describes the goal as getting the labels cheaper, so if all measurement would be available Accuracy would be 100%. This assumption of no measurement error (zero aleatoric uncertainty) is not reasonable in practical biological assays. At first glance reformulating the problem with measurement noise is not trivial. For example, sometimes it may be beneficial to spend the budget on a repeat if the model flags it as “suspicious” (model based on other data consistently predict different value than the measured one)

**On page limit**
The manuscript is over the page limit though not by much, as on page 5 a page break is left in. Please fix this.
also 392-393 line break should be removed

Minor:

Line 402 “We have shown that inference set design that inference set design”

L568-569 “With inference set design, the active agent is again able to outperform thebaselines (see Figure 10). ” <- this should be Figure 9 right?

Define $KL(\hat{\mu}_t, a)$ in eq 5

**Questions:**

1) Can you point out a few practical scenarios where ISD is relevant while Bayesian Optimisation not? (see Weaknesses)
2) How can the method handle measurement error?
3) Were you able to see any meaningful pattern in the hard to predict examples in the public dataset chemical problems? (HOMO-LUMO gap and energy prediction)
4) Pleas fix page limit, and the minor issues.

---

> ### Author Response · Authors · 2024-11-20
> **Author Response to Reviewer #1 (Et7D) -- (1/2)**
>
> We wish to thank Reviewer Et7D for the interesting points highlighted in this review. We have fixed the formatting error and all of the minor issues mentioned in this review. We now aim at answering the additional questions of the reviewer and provide some clarifications regarding both the limitations of the proposed paradigm and the scope of this work.
>
> > 1. Reviewer-Et7D:"The argument by the authors, that the benefit of active learning should be interpreted more broadly if the goal is to label the dataset is valid. Using similar argument, however, most of the time the final goal is to maximize/minimize the outcome(s) of the experiment, the problem become a Bayesian optimization problem, with already existing literature and efficient methods. [...] Can you point out a few practical scenarios where ISD is relevant while Bayesian Optimisation not?"
>
> Indeed, Active Learning and Bayesian Optimization are closely related topics, and we discuss this distinction in our Related Work. Bayesian optimization assumes we have a single known target of interest at the time of running the experiment. That is reasonable if one is trying to find a single drug for a single target, but in industrial drug discovery it is far more cost effective to collect experimental data that allows one to search for many drugs that target many different diseases (including those that might not have been anticipated at the time of the experiment). In this work, we focus not on cases where the experimenter seeks to find samples that maximise one particular evaluation metric, but on problems for which all predictions made by the system are valuable, and can be used for several downstream tasks.
>
> Examples of such cases occur in drug discovery, where the set of samples of interest might be a finite screening library of some tens or hundreds of thousands compounds. In a drug discovery program, we might seek a compound that yields a similar effect as deactivating GeneA (target gene), but importantly, that also does not perturb the functioning of GeneB (specific off-target effect we wish to avoid). These two desiderata *could* in principle be combined into a single reward function and used as objective to maximise for a bayesian optimization algorithm, but doing so would limit the usability of the data collection process to only this program. The potential for biological maps (see Section 4.3, updated manuscript) is to produce generic, reusable relatable biological datasets across large numbers of pre-clinical programs. This goal is best achieved by producing generic predictions about each input (without attempting to maximize or minimize these metrics) and leave it to the downstream processes to use that data for different applications. Such an approach is best served by active learning methods than bayesian optimization.
>
> > 2. Reviewer-Et7D:"if all measurement would be available Accuracy would be 100%. This assumption of no measurement error (zero aleatoric uncertainty) is not reasonable in practical biological assays. At first glance reformulating the problem with measurement noise is not trivial. For example, sometimes it may be beneficial to spend the budget on a repeat if the model flags it as “suspicious”. [...] How can the method handle measurement error?"
>
> This is an important observation and constitues a limitation of the method in its current form. With the paradigm of hybrid screens and inference set design, we present a framework for obtaining a similar performance on a target set of interest without having to obtain the labels for all the samples in the target set. The quality of the hybrid dataset generated from the combination of collected labels and model predictions is inherently upper-bounded by the quality of the dataset as a whole if it was acquired completely. Incorporating measurement error, which could lead to better allocation of resources, is an important aspect of successfully deploying active learning solutions for biological data collection, but this consideration is somewhat perpendicular to the purpose and claims made in this work and we leave the integration and direct modeling of aleatoric uncertainty in inference set design for future work.
>
> We now discuss this consideration in the revised manuscript (see Section 5, updated manuscript).

---

> ### Author Response · Authors · 2024-11-20
> **Author Response to Reviewer #1 (Et7D) -- (2/2)**
>
> > 3. Reviewer-Et7D:"Were you able to see any meaningful pattern in the hard to predict examples in the public dataset chemical problems?""
>
> We looked into this question in some more detail. For the QM9 dataset, we observed that hard to predict examples have higher than average synthetic accessibility (SA) score, which suggests that these molecules can be considered more complex (see Figure 6, updated manuscript). While this represents an interesting question that could motivate a much broader analysis, one of the strengths of active learning approaches is that this notion of "harder examples", which is model-dependent, can be abstracted away from the experimenter and used as a powerful criterion for acquisition ordering without being constrained to concepts that must be easily interpretable by the experimenter. For this reason, we do not recommend using active learning for the sole purpose of identifying a potential heuristic-based ordering of the samples, but to truly leave it to the model to decide which samples to prioritise, without too much human intervention.
>
> > 4. Reviewer-Et7D:"L568-569 “With inference set design, the active agent is again able to outperform the baselines (see Figure 10). ” <- this should be Figure 9 right?"
>
> To clarify, this sentence in the original submission was pointing to the correct figure showing results on our proprietary dataset. The results for RxRx3 are presented in Appendix (see Appendix E.2, updated manuscript).

---

> > ### Comment · Reviewer_Et7D · 2024-11-25
> > **Response to Authtor's comments**
> >
> > Thank you for the detailed answers.
> > Your argument on the late stage optimization is totally clear, and indeed in multi-objective setup this is a rational choice.
> > My question about identifiable patters was purely motivated by scientific interest and I was not suggesting to use these patterns for selection inside an acquisition loop. It can be useful however to understand what contribute to difficulties for a given class of models, or type of descriptors (outside of the scope of present work).

---

> > > ### Author Response · Authors · 2024-11-25
> > > **Author response**
> > >
> > > ### A quick follow-up
> > >
> > > > My question about identifiable patters was purely motivated by scientific interest and I was not suggesting to use these patterns for selection inside an acquisition loop. It can be useful however to understand what contribute to difficulties for a given class of models, or type of descriptors (outside of the scope of present work).
> > >
> > > Indeed! We strongly agree that on top of their ability to reduce experimental costs and accelerating large molecular space exploration, analysing the patterns identified and exploited by an active learning agent represents one of the most interesting potential benefits of such systems. Through reverse-engineering of the sampling behaviours of the agent, we believe that successful deployments of active learning in complex industries such as drug discovery can constitute real opportunities for ML-based solutions to yield scientific insights.
> > >
> > > ---
> > >
> > > ### Zooming out: summary of contributions
> > >
> > > Since our last response, we have uploaded two additional revisions of the paper. We would like to take this opportunity to briefly reiterate our main contributions:
> > >
> > > **1. Introduction of Inference Set Design:** We introduce a novel active learning paradigm called inference set design for hybrid screens, which allows for significant reductions in experimental costs without relying solely on generalization improvements.
> > >
> > > **2. Principled Stopping Criterion:** We propose a principled stopping criterion applicable to both confidence-based acquisition functions and random samplers, with detailed derivations and proofs provided in Appendix A.
> > >
> > > **3. Extensive Experimental Validation:** We validate our claims experimentally by comparing 9 acquisition functions across 7 datasets, including both classification and regression tasks. We believe the results are robust and compelling, demonstrating that inference set design consistently outperforms generalization-based active learning. Additionally, we include experiments on a challenging real-world phenomics dataset used in drug discovery.
> > >
> > > **4. Validation of Theoretical Elements:** We experimentally validate both the assumption of Lemma 1 (Appendix F.1) and the probabilistic bound used in our stopping criterion (Appendix F.2).
> > >
> > > ---
> > >
> > > Thank you once again for your thoughtful review and engagement in this discussion. We believe that this work presents a clear & novel departure from prior methods, and want to make sure this is clear in the revised manuscript. Did you have any remaining concerns or questions that we can address before the end of the discussion period?

---

### Author Response · Authors · 2024-11-20
**General Author Response**

We sincerely thank the reviewers for their time and effort in providing feedback and constructive criticism on our work. In our individual responses to each reviewer, we address their specific comments and questions. In our global response below we highlight commonalities in the reviews and report the most important changes made to the manuscript.

All reviewers have commented on the relevance of the real-world challenges that are at the center of this work, on the pacticality of the proposed solution and on the breadth of tasks and datasets covered in the empirical evaluation. Reviewers have jointly raised the following concerns:
- A page formatting error in the submitted manuscript.
- A lack of additional acquisition functions used as baselines.
- A lack of clarity and important typos in the theoretical section.
- The absence of a link to the code.

Here is a summary of the changes that we made to our submission:
- We have corrected the formatting error causing a blank page in the initial submission.
- We have re-run several experiments with additional acquisition functions and adapted the main text to discuss these results (see Sections 4.2 and 4.3, updated manuscript).
- We have clarified the theoretical section and rectified the notation errors pointed out by Reviewers \#3 and \#4 (see Section 2.2). We have also added a detailed derivation of both Lemma 1 and the bound constituting our stopping criterion in Appendix (see Appendices A.1 and A.2, updated manuscript). Note that the essense of our theoretical contribution remains exactly the same, but its presentation has been improved.
- We added an empirical validation of the assumption made in Lemma 1 and of the correctness of the bound constituting our stopping criterion (see Section 4.4 and Appendix F, updated manuscript).
- We have made available to the reviewers an annonymized version of our code: https://anonymous.4open.science/r/inference_set_design-889D

**Note:** we have highlighted in green the added text or significantly modified paragraphs in the updated manuscript to help reviewers identify the most relevant sections to revisit. Minor modifications for space optimisation and clarity were also realized in other parts of the manuscript.

We thank the reviewers once again for their time in reviewing our revision, and we remain available for any additional questions or concerns!

---

### Author Response · Authors · 2024-12-02
**Authors' closing remark**

We would like to thank all reviewers for what has been a pleasant and productive revision cycle for this manuscript. We highlight below the most important changes made to the paper during the last two weeks:

- We have rewritten and detailed the derivation and proof of Lemma 1 and the rest of our theoretical section for enhanced clarity (see Section 2.2 & Appendix A).
- We have ran additional experiments to add 2 active learning baselines (BALD and Diversity-based agents) as well as 3 heuristic orderings to more completely support the proposed paradigm.
- We have added extensive experiments for validating our weak calibration assumption and proposed stopping criterion (see Appendix F).
- We have made our code available to the reviewers at: https://anonymous.4open.science/r/inference_set_design-889D

In addition to these major changes, we have addressed all minor concerns from the reviewers and actively participated in the discussion to address their questions.

We appreciate the thoughful input and believe these changes adequately address the reviewers' comments and significantly increased the quality of the paper.

---

### Meta-Review · Area_Chair_8kQc · 2024-12-21

**Metareview:**

The paper proposes a subset selection method for biological data labeling to achieve high accuracies with few examples. The reviewers are largely in agreement that this is an important problem, and the application area is novel with real world relevance. There were some concerns about novelty of the proposed method, its technical depth being limited and need for further empirical validation.

**Additional Comments On Reviewer Discussion:**

The reviewer discussion largely consisted of a few technical clarifications and better understanding of the technical setup and data details. The reviewer-authors exchanges were very productive and several reviewers raised their scores by quite bit.

---

### Decision · Program_Chairs · 2025-01-22

Accept (Poster)